# Aptamer-Based Recognition of Breast Tumor Cells: A New Era for Breast Cancer Diagnosis

**DOI:** 10.3390/ijms25020840

**Published:** 2024-01-10

**Authors:** Natassia Silva de Araújo, Aline dos Santos Moreira, Rayane da Silva Abreu, Valdemir Vargas Junior, Deborah Antunes, Julia Badaró Mendonça, Tayanne Felippe Sassaro, Arnon Dias Jurberg, Rafaella Ferreira-Reis, Nina Carrossini Bastos, Priscila Valverde Fernandes, Ana Carolina Ramos Guimarães, Wim Maurits Sylvain Degrave, Tatiana Martins Tilli, Mariana Caldas Waghabi

**Affiliations:** 1Laboratório de Genômica Aplicada e Bioinovações, Instituto Oswaldo Cruz, Fiocruz, Rio de Janeiro 21040-900, RJ, Brazil; natassiasaraujo@gmail.com (N.S.d.A.); amoreira@ioc.fiocruz.br (A.d.S.M.); rayane.abreu04@gmail.com (R.d.S.A.); vargasjuniorvc@gmail.com (V.V.J.); deborah.antuness@gmail.com (D.A.); juliabadarom@hotmail.com (J.B.M.); tayannefelippe.fiocruz@gmail.com (T.F.S.); carolg@fiocruz.br (A.C.R.G.); wim.degrave@fiocruz.br (W.M.S.D.); 2Laboratório de Pesquisas sobre o Timo, Instituto Oswaldo Cruz, Fiocruz, Rio de Janeiro 21040-900, RJ, Brazil; ajurberg@gmail.com (A.D.J.); rafinha_frbe_4@hotmail.com (R.F.-R.); 3Laboratório de Animais Transgênicos, Universidade Federal do Rio de Janeiro (UFRJ), Rio de Janeiro 21941-902, RJ, Brazil; 4Instituto de Educação Médica (IDOMED), Universidade Estácio de Sá (UNESA)—Campus Vista Carioca, Rio de Janeiro 20071-004, RJ, Brazil; 5Divisão de Patologia (DIPAT), Instituto Nacional do Câncer (INCA), Rio de Janeiro 20230-130, RJ, Brazil; nina.bastos@inca.gov.br (N.C.B.); pvalverde@inca.gov.br (P.V.F.); 6Laboratório de Fisiopatologia Clínica e Experimental, Instituto Oswaldo Cruz, Fiocruz, Rio de Janeiro 21040-900, RJ, Brazil; tatiana.tilli@fiocruz.br; 7Plataforma de Oncologia Translacional, Centro de Desenvolvimento Tecnológico em Saúde, Fiocruz, Rio de Janeiro 21040-900, RJ, Brazil

**Keywords:** breast cancer, aptamers, diagnosis, computational modeling

## Abstract

Breast cancer is one of the leading causes of death among women worldwide and can be classified into four major distinct molecular subtypes based on the expression of specific receptors. Despite significant advances, the lack of biomarkers for detailed diagnosis and prognosis remains a major challenge in the field of oncology. This study aimed to identify short single-stranded oligonucleotides known as aptamers to improve breast cancer diagnosis. The Cell-SELEX technique was used to select aptamers specific to the MDA-MB-231 tumor cell line. After selection, five aptamers demonstrated specific recognition for tumor breast cell lines and no binding to non-tumor breast cells. Validation of aptamer specificity revealed recognition of primary and metastatic tumors of all subtypes. In particular, AptaB4 and AptaB5 showed greater recognition of primary tumors and metastatic tissue, respectively. Finally, a computational biology approach was used to identify potential aptamer targets, which indicated that CSKP could interact with AptaB4. These results suggest that aptamers are promising in breast cancer diagnosis and treatment due to their specificity and selectivity.

## 1. Introduction

Breast cancer is the most common cancer among women worldwide and is among the leading causes of death among the female population. According to the World Health Organization (WHO), 2.9 million cases were reported worldwide in the year 2018 [1,2]. Breast carcinoma is a complex disease, and its molecular diagnosis and therapeutic indication are based on histopathological findings from tissue biopsy samples in which the presence and expression levels of estrogen and progesterone hormone receptors (ER and PR) and growth receptors (HER2) in the membrane of the tumor cells are investigated [3]. An analysis of these receptors indicates which target therapy should be applied for each patient, as an initial attempt to personalize therapy. The tumors that are estrogen-negative, progesterone-negative, and HER2-negative are known as triple-negative (TN) tumors, accounting for about 15% of breast tumors, and are considered to present worse prognoses. In this case, the patient will not benefit from target therapy, and chemotherapy is the mainstay of treatment. Although molecular diagnosis has provided great progress in the fight against breast tumors, the disease still presents important challenges. In search of greater specificity for the diagnosis of breast cancer, precision medicine has invested in tools that can act as biosensors for the identification of tumor-specific molecules, among them are the aptamers [4]. 

Aptamers are synthetic oligonucleotides of DNA or RNA formed by a single-stranded sequence flanked by two constant regions, which allows for synthesis using PCR amplification. According to their primary sequences, aptamers acquire unique 3D structures, thus being able to recognize with high selectivity and affinity a wide range of molecules with high biological relevance [5,6,7]. The selection of aptamers could be performed with a library of random sequences and using the systematic evolution of ligands by exponential enrichment (SELEX) technique [8]. The goal is to select and identify specific aptamer sequences for target molecules by exposing these targets to an aptamer’s sequences repeatedly. As a variation of the technology of aptamer selection, the CELL-SELEX method stands out as an important tool [9,10]; in this case, aptamers are incubated with a tumor cell and a non-tumor control cell, thus allowing the selection of aptamers that specifically recognize tumor cells [11,12]. Aptamers are often compared to antibodies due to their similarity in applications. However, the process of obtaining and improving aptamers is much easier and less costly compared with monoclonal antibody production. The application of aptamers in the diagnostic field is diverse and can be used in techniques such as histochemistry and even for liquid biopsy. Thus, aptamers are presented as biosensors for bio-recognition with possible application for molecular diagnosis [13,14]. Some groups are already studying the application of multiplex aptamer–histofluorescence using more than one aptamer as a molecular probe to characterize the profile of heterogeneity in glioblastoma [15]. In the present study, we propose the identification of tumor-specific aptamers that can be used as a diagnostic and prognostic tool for breast cancer. Here, we selected aptamers for breast tumor cell lineage that are capable of recognizing subpopulations of the triple-negative subtype. Furthermore, we validated the recognition in a 3D model and subsequently in primary and metastatic site tumor samples. The Cell-SELEX approach has advantages in selecting aptamers against living cells as it preserves proteins in their natural conformation. However, it is not possible to conclude which target the aptamer recognizes. Therefore, we used in silico methods to describe possible aptamer targets by predicting the structure of the aptamer and its possible cellular target to perform molecular docking, potentially leading to the discovery of new tumor biomarkers. Finally, we believe that aptamers can play an additive role in conventional molecular diagnostics as well as being an important tool for early metastasis detection.

## 2. Results

### 2.1. Selected Aptamers Specifically Recognize Breast Tumor Cells

To evaluate the tumor cell detection ability of each of the aptamers, we used increasing concentrations of FAM-conjugated aptamers and flow cytometry. In general, all five aptamers (AptaB1, AptaB2, AptaB3, AptaB4, and AptaB5) exhibited a dose response in cellular recognition, ranging from 9% at the lowest dose of 25 nM to 96% at the highest dose of 400 nM (Figure 1). The incubation of tumor cells with FAM-conjugated aptamers at the higher dose of 400 nM presented the best results: MDA-MB-231 cell recognition varied from 60% (AptaB1) to 75% (AptaB2), 96% (AptaB3), 59% (AptaB4), and 94% (AptaB5) (Figure 1A,B). The binding specificity of all five aptamers for breast tumor cell recognition was tested by incubating with non-tumor breast cells, MCF10A, at the same conditions. The results indicated low recognition of non-tumor control cells by AptaB1, AptaB4, and AptaB5, reaching a maximum of 11% of cellular binding with the highest aptamer concentration. However, both aptamers AptaB2 and AptaB3 showed undesired binding to MCF-10A cells, with 50% and 30% of recognition, respectively (Figure 2A,B). To evaluate the binding affinity of aptamers to their target cells, the dissociation constant (Kd) was calculated from the MFI obtained using flow cytometry. The Kd analysis showed values in the nanomolar range, evidencing the good affinity of the aptamers to their cellular targets: AptaB1 presented Kd = 139 nM, AptaB2 presented Kd = 206 nM, AptaB3 presented Kd = 145 nM, AptaB4 presented Kd = 194 nM, and AptaB5 presented Kd = 126 nM (Table 1).

### 2.2. Subcellular Localization of Aptamers in MDA-MB-231 Cells

To validate the recognition and specificity of the aptamers, we used the aptaflurescence technique, in which the aptamers were incubated with MDA-MB-231 and MCF-10A, and the number of recognized cells was quantified. The MDA-MB-231 cells were recognized by all five aptamers (Figure 3A). The subcellular localization of the aptamers was distributed as follows: AptaB1 labeling was predominantly in the nucleus, as 85% was observed in the nucleus only, 12% in the nucleus and cytoplasm, and 3% in the cytoplasm only. The AptaB2 intracellular localization was distributed as 40% in the nucleus and cytoplasm, 32% in the nucleus only, and 27% in the cytoplasm only. AptaB3 was observed at 42% in the nucleus and cytoplasm, 37% in the nucleus only, and 21% in the cytoplasm only. AptaB4 was observed at 47% in the nucleus and cytoplasm, 34% in the nucleus only, and 18% in the cytoplasm only. Finally, AptaB5 labeling was observed at 55% in the nucleus, 27% in the nucleus and cytoplasm, and 17% in the cytoplasm only (Figure 3A,B). In addition, the cytoplasmic labeling of AptaB2, AptaB3, and AptaB4, presented a punctual profile. Moreover, we observed the absence of staining in the non-tumoral MCF-10A cells (Figure 3B).

Further, to validate the recognition and specificity of the aptamers for the tumor lineage, we used a more complex culture system, MDA-MB-231 spheroids, which allows better cell–cell interactions and mimics the tissue architecture found in the tumor. For this purpose, MDA-MB-231 spheroids were incubated with the five individual aptamers at 400 nM, and we observed that all aptamers recognized the tumor 3D culture (Figure 4B–F). Also, to assess non-specific labeling, the aptamer initial library was incubated with the tumor spheroids, and no labeling was observed (Figure 4A). In parallel, to validate the specificity of aptamer recognition for tumor cells only, we also constructed a 3D model of the non-tumoral control cell line MCF-10A, and an absence or very weak labeling was observed (Figure 4B–F). 

### 2.3. Aptamers Detect an Expanded Panel of Triple-Negative Breast Tumor Cells 

Breast cancer is a highly heterogeneous and diverse disease, and based on this scenario, we evaluated if the aptamers could also recognize other triple-negative breast tumor cells beyond MDA-MB-231, in which the selection process was conducted. The MDA-MB-231 cells lack hormone receptor expression and HER2 overexpression. They are therefore classified as triple-negative. This subtype is the most challenging among the molecular subtypes of breast disease, as it exhibits high complexity and molecular heterogeneity, even being subdivided into intrinsic subtypes according to cell profile. Therefore, to evaluate the potential of the selected aptamers as a diagnostic tool, we investigated the ability of the five aptamers to detect cells of different intrinsic triple-negative subtypes. For this purpose, we performed the aptafluorescence technique, in which cells were incubated with the individual aptamers or with the aptamer initial library at 400 nM. It was observed that all five aptamers showed expressive recognition for the MDA-MB-468 and HCC-70 cell lines: AptaB1, AptaB2, and AptaB5 recognized more cells compared with AptaB3 and AptaB4 (Figure 5). However, none of the tested aptamers recognized the HCC-1937 cell line (Figure 5).

### 2.4. The Aptamer Panel Detects Breast Tumor Cells from the Luminal A, Luminal B, and HER 2 Molecular Subtypes

In addition to the triple-negative cell lines, it would be interesting to investigate whether the five selected aptamers could recognize other breast tumor cells from the other molecular subtypes luminal A, luminal B, and HER2. For this purpose, the FAM-conjugated aptamers were incubated individually with the MCF-7, BT-474, and HCC-1954 cell lines, representing the luminal A, luminal B, and HER2 subtypes, respectively. The aptamers recognized all breast tumor molecular subtypes. We observed a variation in the recognition of the MCF-7 and HCC-1954 cell lines among the aptamers: AptaB1 and AptaB3 showed greater recognition of MCF-7, and AptaB2, AptaB4, and AptaB5 showed equivalent recognition among them (Figure 6).

### 2.5. Aptamers Detect Human Breast Tumor Clinical Samples from Different Subtypes

After characterizing the specificity of the selected aptamers with potential use for breast cancer diagnosis, we next aimed to validate their specificity and application in tissue samples from patients with breast tumors. Thus, tissue microarrays were obtained to test aptamer specificity. Tissue microarrays are paraffin blocks produced by extracting cylindrical tissue cores from different paraffin donor blocks and re-embedding these into a single block, the so-called microarray. Using this technique, up to 100 tissue samples were arrayed for aptamer binding in a single paraffin block. Human breast tumor and non-tumor samples were used to validate the binding capacity and specificity of the selected aptamers. A cohort of 100 breast tissue samples, was distributed as follows: 10 adjacent non-tumoral breast tissues, 40 metastatic carcinomas (lymph node metastatic), and 50 invasive breast carcinoma primary sites (46 invasive carcinomas of no special type, 1 neuroendocrine carcinoma, and 3 medullary carcinoma). 

The individual aptamers were incubated with the TMA slide and the number of recognized samples and staining intensity were analyzed, in which (+) was considered low intensity, (++) was considered moderate intensity, and (+++) was considered high intensity (Figure 7 and Table 2). AptaB1 recognized 14% of invasive carcinoma samples and 12.5% of metastatic tissue samples at high intensity and very low staining intensity in 10% of control adjacent tissue samples (Table 2). AptaB2 recognized 36% of invasive carcinoma samples and 12% of lymph node metastatic carcinoma tissue samples with high intensity and recognized just 10% of adjacent tissue with low intensity. In contrast, AptaB4 and AptaB5 detected 50% and 40%, respectively, of invasive carcinomas with high staining intensity and 37.5% and 67.5%, respectively, of metastatic samples with high labeling intensity, respectively. However, low-intensity labeling in 40% and 30% detection of adjacent tissues was also observed (Table 2). 

In addition, aptamer recognition was also performed to discriminate breast cancer molecular subtypes. We observed that the aptamers recognized the luminal subtype with high intensity: AptaB1 (10%), AptaB2 (28%), AptaB4 (42%), and AptaB5 (32%), except for AptaB3, which recognized just one sample (Figure 7, Table 3). The detection rates of the HER2 subtype were more expressive: AptaB1 (42%), AptaB2 (70%), AptaB3 (14%), AptaB4 (85%), and AptaB5 (70%) (Figure 7, Table 3). The aptamers also detected the triple-negative subtype samples, in which we observed higher detection by AptaB5, AptaB4, and AptaB2 with 50%, 40%, and 10%, respectively, at high intensity (Figure 7 and Table 3). 

Furthermore, the recognition was assessed based on the molecular subtype identified in the metastatic lymph node tissue. Of the 40 analyzed samples, 33 contained data on the expression of classical markers (ER, PR, HER2). This result gave us the opportunity to investigate whether aptamers had differential recognition capabilities in metastatic lymph node tissues according to molecular subtypes. In particular, AptaB1 showed exclusive recognition of metastatic samples classified as luminal subtypes. In contrast, the other aptamers showed recognition in metastatic tissues of all molecular subtypes (Table 4).

To investigate the extent of aptamer recognition, we established a correlation between the binding results observed in the TMA samples and clinical and pathological information. AptaB2, AptaB4, and AptaB5 recognized stage I tumor samples (Table 5). In contrast, all aptamers were able to detect stage II tumors, with AptaB4 and AptaB2 showing the highest sensitivity. Extending our analysis to stage III tumors, we found that all five aptamers showed recognition potential, with the best results for AptaB5, AptaB1, and AptaB4, reaching more than 50% recognition, despite the low number of samples from those stages (Table 5). In addition, a difference in detection was observed between the aptamers according to histological grade. The results showed that the AptaB1, AptaB2, AptaB4, and AptaB5 aptamers recognized at least one Grade I specimen, with AptaB4 and AptaB5 recognizing most of the cases, and these aptamers were also the most efficient in recognizing samples from Grades II and III. The AptaB3 aptamer recognized only Grade II samples (Table 6). 

In addition, we performed an analysis of the recognition of clinical samples by aptamers according to TMN classification. The classification system uses T for tumor size, N for lymph node involvement, and M for metastases in distant organs. This classification is fundamental in oncology for determining the stage of the disease and subsequent therapeutic approaches. Our results showed that AptaB2, AptaB4, and AptaB5 detected samples at earlier (T1N0M0) and more advanced (T4N1M0) stages of the disease. Notably, AptaB5 showed recognition for primary tumor samples with lymph node involvement (T4N1M0) (Table 7).

### 2.6. Evaluation of the Aptamer Panel as a Diagnostic Tool

The calculation of diagnostic indices is a widely used tool for evaluating the diagnostic efficacy of a given biomarker. After quantifying the number of samples recognized by the set of aptamers, it was possible to obtain the parameters of sensitivity, specificity, and accuracy of the test. First, we evaluated the performance of the aptamers individually and observed that the aptamers AptaB1, AptaB2, and AptaB3 showed low sensitivity, with values between 10% and 26%, but high specificity around 90%. The aptamers AptaB4 and AptaB5 had a sensitivity of 44% and 52% and a specificity of 60% and 70%, respectively. The accuracy test showed results that ranged from 21% to 54%. It was then observed that the aptamers recognized different sample profiles, which could allow for greater diagnostic coverage. Finally, we evaluated the combinations of two or more aptamers for diagnostic purposes. The sample recognition sensitivity observed with the combination of the aptamers AptaB4 and AptaB5 was 77%. The combination of the five aptamers AptaB1, AptaB2, AptaB3, AptaB4, and AptaB5 reached 96% sensitivity (Table 8). In addition, given the lack of specific biomarkers for the triple-negative molecular subtype diagnosis, we investigated the recognition profile with the aptamer combinations in this molecular subtype sample. The individual aptamer results showed specificity above 50% and sensitivity ranging from 10% with AptaB1 to 50% with AptaB5. The combinations increased the sensitivity by up to 96% and the specificity by between 30% and 40% (Table 8).

### 2.7. In Silico Characterization of the Three-Dimensional Structure of Aptamers and Selection of Potential Recognition Targets

After studying aptamer recognition in various cell lines and tissues, our goal was to elucidate the potential molecular targets recognized by these aptamers. We first attempted to elucidate the structure of the aptamers and then investigated the possible targets of the aptamers based on proteomic data from the literature on the target tumor cell line MDA-MB-231 and the control non-tumor cell line MCF-10A [16,17]. The sequential steps of the aptamer analysis and potential target protein analysis are described in a pipeline (Appendix A). 

### 2.8. Three-Dimensional Structural Characterization of Selected Aptamers for MDA-MB-231 Cells 

The 2D structures predicted with NUPACK and mFold showed some general similarities, such as large loops and minimal internal base pairing, suggesting open, flexible conformations. However, the specific motifs differed, as evidenced by the variations in free energy values calculated using the two servers.

For most aptamers, the mFold and NUPACK free energy values differed somewhat, indicating differences in the predicted stability of the 2D folds. The one exception was AptaB2, for which both servers calculated identical free energies of −6.15 kcal/mol. This implies that the servers predicted the same minimum free energy 2D structure for AptaB2 (Figure 8).

Beyond the 2D structure, we also wanted to evaluate how the predicted 3D conformations compared between mFold and NUPACK. To make this comparison, we used RNA-align to calculate the TM score between the 3D models. The TM scores were low (<0.3) for most of the aptamers, indicating substantial 3D structural deviations between the mFold and NUPACK models. However, AptaB2 had a high TM score of 0.867, indicating strong 3D structural similarity (Figure 9).

In summary, while NUPACK and mFold predicted similar general 2D features, the exact base pairing and resulting 3D structures differed for most aptamers. However, AptaB2 showed consistency between servers for both 2D and 3D conformations, likely due to its inherent structural rigidity.

Since the NUPACK models showed lower minimum free energies for the predicted 2D structures, as well as greater 3D structural folding, these were selected as the starting 3D conformations for further refinement using molecular dynamics simulations. By starting with the NUPACK models, which appeared to be in lower energy states, the simulations could provide deeper insights into the structural dynamics and inherent flexibility of these aptamers.

### 2.9. Aptamer Structures Are Stable in Aqueous Solution

The structures generated with the NUPACK program with the lowest free energy ΔG values went on to the molecular dynamics stage. Molecular dynamics simulations revealed that the structures of the five aptamers varied greatly from the conformations obtained after the equilibration step. The RMSD of all systems raised over 13 Å in the first 50 ns of the simulation. The RMSD of AptaB1 and AptaB2 raised gradually until 15 Å and showed huge oscillations, ranging from 6 to 18 Å. AptaB3 was the most stable aptamer since its RMSD was smoothly elevated over the simulation. AptaB4 reached near 28 Å and stabilized around 25 Å after the first 250 ns. Finally, AptaB5 fluctuated from 8 to 15 Å for most of the simulation (Figure 10).

A cluster analysis was carried out to group the conformations assumed by each aptamer according to the RMSD variation. Several clusters were formed for each of them. AptaB2 formed 19 groups of conformations and was the system with the lowest number of clusters, whereas AptaB4 reached 45 clusters. Figure 11 shows the five biggest clusters observed for each aptamer. In general, most of the aptamers formed a major group of conformations encompassing over 50% of the simulation range. However, for AptaB5, we observed two major clusters, indicating the existence of two stable conformations (Figure 11E). In this context, we retrieved one representative structure for each aptamer, except AptaB5, for which we considered two representative structures for the molecular docking steps.

### 2.10. Search for Potential Aptamer Targets in MDA-MB-231 Cell Lines

To identify the potential protein targets recognized by the aptamers within the lineage from which they were selected, we used proteomic data obtained from both the MDA-MB-231 and non-tumor MCF-10A cells, as provided by Lawrence et al., 2015 [16]. We specifically selected 289 proteins that exhibited at least a 2-fold increase in the target lineage compared with the control lineage. We then confirmed the overexpression of the selected proteins using a second study performed by Ziegler et al., 2014 [17] which specifically examined the proteomics of membrane proteins present in the cells of interest. Following these criteria, we identified 40 candidate proteins for the subsequent molecular docking phase (Appendix A). 

Taking into account our aptafluorescence results, we found that the aptamers initially selected for the MDA-MB-231 cell line also recognized the MDA-MB-468, BT-474, and MCF-7 cell lines but did not recognize the HCC-1937 cell line. We evaluated the data from Lawrence et al., of which the forty selected proteins were overexpressed in the MDA-MB-468, BT-474, and MCF-7 cell lines and under-expressed in the HCC-1937 cell line. Using this filter, we identified four proteins: CSKP, TMEM205, CD151, and TM9S3. The proteins selected as possible targets were evaluated for their electrostatic potential. This information is important because positively charged amino acids favor the binding of negatively charged nucleic acids. We observed that the four selected proteins presented spatial orientation with large portions of their structure oriented toward the extracellular region of the lipid bilayer with positively charged regions (in blue). This indicates electrostatic potential for interacting with the negatively charged DNA aptamers (Appendix A), and thus validation to proceed to the next step of molecular docking. 

### 2.11. Characterizing Protein–Aptamer Complexes using Molecular Docking 

For the molecular docking stage, we utilized the four chosen proteins (CSKP, TMEM205, TM9S3, and CD-151) and the central structures of the largest cluster that was observed with molecular dynamics of the aptamers individually (AptaB1, Cluster 1, AptaB2, Cluster 1, AptaB3, Cluster 1, AptaB4, Cluster 1, AptaB5, Cluster 1 and Cluster 2). Following the molecular docking process, a more negative Haddock score indicates a higher recognition of spontaneity. Hence, we selected the best complexes based on the score values observed (see Table 9). The docking results reveal the highest scores for the TM9S3–AptaB1 (Haddock score = −76.7), CD151–AptaB2 (Haddock score = −53.1), TMEM205–AptaB3 (Haddock score = −34.2), CSKP–AptaB4 (Haddock score = −34.7), TM9S3–AptaB5.1 (Haddock score = −41.2), and TM9S3–AptaB5.2 (Haddock score = −81.2) complexes. After coupling, the complex file was utilized as input in the OPM to display the outcome in a system with membranes (Figure 12). 

Following the molecular docking step, we obtained six complexes, one for each of the four first aptamers (AptaB1–AptaB4) and two for AptaB5, which had two conformations (AptaB5.1 and AptaB5.2) submitted to molecular docking, hence producing two independent outcomes. The complex selection was performed in a way that each aptamer was complexed to a protein, and we selected the conformations that generated the most negative Haddock scores. 

### 2.12. Molecular Dynamics Details the Interactions between Proteins and Aptamers

The hydrogen bond (hbond) occupancy between two pairs of atoms is one of the most important measurements to study interactions using molecular dynamics. In this regard, hydrogen bonds were formed between the aptamer and the protein in all evaluated complexes. In general, each system had at least four pairs of atoms forming hydrogen bonds with occupancies higher than 20% of the simulation time (Figure 13). In particular, the complex composed of CSKP and Apta4 stood out in terms of the number of interactions, as well as in hbond occupancy. In this context, CSKP^Apta4^ had over 30 pairs of atoms forming hbonds with occupancies over 20% and 12 hbonds that remained for at least 95% of the simulation time (Figure 13D). Contrastingly, the complex formed by CD151 and Apta2 showed lower hbond occupancies among all systems. The higher hbond occupancy for this system was between GLN 194 and C 322, which was kept for around 30% (Figure 13B). Interestingly, the systems TM9S3^Apta5.1^ and TM9S3^Apta5.2^ are composed of the same protein and the same aptamer sequence, diverging only by the conformation of the aptamer. Even so, the hbond interactions formed in those two complexes were not identical. Both systems formed four hbonds with occupancies over 20% and had ASN 53 as the residue involved in the most stable hbond with nearly 74% occupancy. However, TM9S3^Apta5.1^ formed an hbond with the aptamer involving the residue HID 127 that did not appear among the interactions over 20% occupancy in TM9S3^Apta5.2^. Similarly, the pair formed by C 617 and LYS 542 in TM9S3^Apta5.2^ was not among the most stable hbonds of TM9S3^Apta5.1^. 

Following the hbond calculations, we applied the MM/GBSA method to estimate the ΔG_bind_ between each aptamer/protein complex and assess the interactions from the energy perspective. ΔG_bind_ estimations are among some of the most reliable indications of the interaction between two molecules and provide information about the spontaneity of the recognition process.

Overall, evaluations on all systems resulted in negative ΔG_bind_ values, indicating molecular association (Table 10). The most negative ΔG_bind_ was observed for the system composed of CSKP and AptaB4 (−117.7 ± 2.2 kcal/mol), followed by the complex formed by TM9S3 with AptaB1 (−68.1 ± 2.4 kcal/mol). Contrastingly, the interaction between CD151 and AptaB2 appears to be the weakest among all systems (−3.6 ± 3.7 kcal/mol). These results suggest that CSKP might be a target for AptaB4. 

### 2.13. Affinity Calculations Suggest Spontaneous Interactions between the Aptamers and the Proteins

ΔG_bind_ estimations are among the most reliable indications of an interaction between two molecules. In this regard, the ΔG_bind_ estimation between the best complex aptamer/protein, as chosen using the molecular docking assay, provided a perspective on the spontaneity in the recognition process through the energetic behavior of the interacting molecules.

Overall, evaluations on all systems resulted in negative ΔG_bind_ values, indicating molecular association. The most negative ΔG_bind_ was observed for the system composed of CSKP and Apta B4 −117.7 ± 0.2 kcal/mol, followed by the complex formed by TM9S3 with AptaB5.2 −98.4 ± 0.5 kcal/mol. Contrastingly, the interaction between TMEM205 and AptaB2 appeared to be the weakest among all systems (Table 10). These results suggest that CSK and TM9S3 might be targets for AptaB4 and AptaB5, respectively. 

All values are given in kcal/mol. ΔG_bind_ = ΔE_vdw_ + ΔE_ele_ + ΔG_esurf_ + ΔG_egb_. The average error follows the symbol “±”. 

We also decomposed the ΔG_bind_ to evaluate the individual contribution of each residue to ΔG_bind_. In general, the residues forming the strongest hbonds in each system also appeared to have the most negative energies. In other words, those residues contributed more to achieving the negative ΔG_bind_ observed and kept the interaction between the protein and the aptamer (Figure 14). Once again, CSKP^Apta4^ was distinguished among the systems in this study by the elevated number of residues showing negative energies (Figure 14D). This finding is in accordance with the elevated number of hbonds previously described for this complex.

## 3. Discussion 

Breast cancer is a serious public health problem, and the aim of the present work was to identify specific aptamers for this tumor type with potential diagnostic function. Currently, the diagnosis of the disease is made by searching for receptors present in the tumor cell membrane (ER, PR, and HER2), which allows the classification of tumors into four major molecular subtypes [18]. Tumors expressing the hormone receptors ER and PR or the growth factor HER2 benefit from targeted therapies, and the triple-negative subtype is treated with cytotoxic chemotherapy and, in some cases, immunotherapy [19,20]. However, in clinical practice, the molecular classification based on these three receptors is not sufficient to define a more accurate diagnosis and prognosis, mainly for the triple-negative subtype [21]. The study of ssDNA aptamers capable of detecting the disease can serve as a tool to complement the currently used diagnosis, and the identification of the aptamer’s targets could be used as new TNBC biomarkers, thus developing strategies for personalized therapy. 

Initially, flow cytometry was used to characterize the specificity of the aptamers in recognizing only tumoral cells. The analyses showed that the five selected aptamers were effective in recognizing MDA-MB-231 tumor cells. A dose response was observed in the detection of tumor lineage by the aptamers. The analyses indicated that AptaB3 and AptaB5 were most effective in detecting the highest number of cells, followed by AptaB2, AptaB4, and AptaB1, respectively. In addition to tumor specificity, the affinity for the recognized target was analyzed using the dissociation constant (Kd), where the lower the value of Kd, the stronger the affinity of the aptamer to its target. The aptamer AptaB5 had the lowest Kd, followed by aptamers AptaB3, AptaB4, and AptaB1. Aptamer AptaB2 showed a very high Kd, indicating a low affinity for its target. The aptamers described in the literature have Kd values ranging from 10 to 800 nM, depending on the recognized target [22], such as the aptamers for the CD63 receptor with a Kd of 100 nM and for CA50 with a Kd of 30.7 nM [23]. With the advancement of aptamer improvement techniques, many authors are adding punctate modifications to aptamers to increase the stability and affinity of the aptamer for its targets, thus presenting lower Kd values. The aptamers used in the present study do not have structural modifications and, therefore, the data are based on the specificity and affinity of the aptamers in their original conformation. The Kd values observed in the assays ranged from 126 nM for the aptamer AptaB5 to 206 nM for AptaB2. 

The identification of five specific aptamers is particularly relevant in the context of intra-tumor heterogeneity, a concept based on the composition of different cellular subpopulations within the tumor mass, and clusters of cells with genetic and phenotypic variations. Its causes and consequences are widely discussed and reviewed in the literature, and it is believed to be the result of the carcinogenesis process, in which the accumulation of random mutations occurs during tumor development, with interference from the microenvironment and the immune system [24]. Although it is a concept more addressed in the clinic, cellular heterogeneity has already been reported in the MDA-MB-231 lineage, and some studies point to the existence of subpopulations within this cell, such as a stem cell-like profile, presenting a higher expression of surface receptors as CD44, CD105, and CD90 [25,26,27]. In addition, breast cancer therapy is still a challenge, as the drugs in use are not specific for different clusters but rather treat the tumor as a homogeneous mass, consequently impairing the total elimination of tumor cells that do not have sensitivity to the drug used, which favors disease relapse [28]. Due to their high specificity, aptamers can identify subtle differences among cells [29], and one of their applications is the targeted delivery of drugs [30]. These two characteristics make them a potential tool to circumvent the problem of tumor heterogeneity; thus, it is extremely important that the aptamer does not have expressive recognition of the non-tumor cell. Therefore, we investigated the recognition and the affinity of the five aptamers by the non-tumor MCF-10A cells, showing low recognition and low affinity, with the exception of AptaB2 and AptaB3, which recognized the non-tumor cell but with low affinity, as indicated by the high Kd values.

Aptafluorescence assays were also performed to evaluate aptamer recognition and specificity, which confirmed that all aptamers recognized the MDA-MB-231 cells. Curiously, with the use of the 2D culture model, no aptamer binding was observed for the non-tumoral breast cell MCF-10A, but the flow cytometry analysis revealed MCF-10A recognition by AptaB2 and AptaB5. Therefore, we could speculate that the trypsinization process performed for the flow cytometry analysis could alter the cell membrane and thus may have caused this inconsistency, as opposed to the adhered cells used in aptafluorescence.

In addition, the aptafluorescence analysis suggested the cellular localization of the aptamers after binding to their targets. It was observed that after 1 h incubation with tumor cells, the aptamers AptaB2, AptaB3, AptaB4, and AptaB5 were predominantly localized in the nucleus and cytoplasm, while aptamer AptaB1 was only localized in the nucleus. This information could help to understand the intracellular tracking of the selected aptamers, giving clues to their biological function and possible mechanisms involved in a potential anti-tumor action.

In addition to the analysis in 2D culture, the use of the aptamers was also evaluated in a three-dimensional (3D) cell culture model. In recent years, several studies have shown how this system overlaps with 2D models, as 3D culture provides greater interaction between cells and more closely mimics the tissue scaffold, representing an important advance in the field of cell biology. Therefore, we investigated if the selected aptamers would be able to recognize tumor cells in a more complex culture model. The results corroborate those obtained in the monolayer model. All five aptamers recognized the tumor cell spheroids (MDA-MB-231), with no recognition of the non-tumor control cell lineage spheroids (MCF-10A). 

After evaluating tumor cell specificity for the MDA-MB-231 cells, the extension of the aptamer’s recognition was explored in other BC subtypes. The TNBC diversity is stratified into intrinsic subtypes according to the molecular profile of the cells. Aptamers were selected for the MDA-MB-231 lineage, which belongs to the intrinsic subset of the mesenchymal type, whose main characteristic is the expression of genes related to cell differentiation and epithelial-mesenchymal transition. The profile of this subset is characterized by cells with higher invasive potential and strong interaction with the extracellular matrix and is reported to be associated with approximately 20% of TNBC cases. To further investigate the ability of aptamers to detect breast tumors from other intrinsic subsets of TNBC cells, we performed an aptafluorescence assay using the MDA-MB-468 and HCC-1937 lines, which belong to the basal-like 1 (BL1) intrinsic subgroup. Interestingly, all five aptamers showed strong recognition for the MDA-MB-468 cell line and did not recognize the HCC-1937 cell line. Although they belong to the same intrinsic subtype, they are molecularly and genetically distinct. The HCC-1937 cell line was isolated from a primary tumor of germline origin in a premenopausal woman. The MDA-MB-468 cell line was isolated from a pleural infiltrate of a postmenopausal woman, as well as the MDA-MB-231 cell line [31]. The BL1 and BL2 subtypes account for approximately 50% of TNBC. Therefore, we also evaluated aptamer recognition for the HCC-70 lineage, which belongs to the intrinsic BL2 subtype. Moderate aptamer recognition of this lineage was observed. The BL1 and BL2 intrinsic subtypes share basal characteristics but were divided into two subtypes because BL1 is characterized by higher expression of genes involved in DNA repair and cell cycle progression, while BL2 is characterized by higher expression of growth factor-related genes. Our results thus demonstrate that aptamers could recognize cells with different molecular profiles, including cells of the three most common intrinsic subtypes in TNBC. These data are particularly relevant because it is worth noting that TN is diagnosed based on the absence of expression of known classical receptors (ER, PR, and HER2). Therefore, it is of great value to identify tools capable of precisely detecting this subtype as additional diagnostic information. After verifying that the aptamers used in the present study recognized other TN cells in addition to the lineage in which they were selected, we also evaluated their possible use to recognize lineages of other subtypes of BC, as an interesting analysis to evaluate the specificity of aptamers for the TN subtype. For this purpose, we also performed an aptafluorescence assay with luminal A (MCF-7), luminal B (BT-474), and HER2 (HCC-1954) lines and observed that the five aptamers recognized lines of all molecular subtypes, expanding their applicability in tumor diagnosis. 

Finally, the validation step was performed on breast cancer tissue samples, using the TMA assay, in which we confirmed the detection of tumor tissues of all molecular subtypes. Interestingly, we highlighted that AptaB4 recognized tumors from initial phases such as stage I, Grade I, and TNM classification T1 and T2, indicating its potential use for the early detection of BC. Furthermore, we performed a comparison of aptamer recognition on invasive carcinoma samples, lymph node metastatic carcinoma samples, and tissue samples adjacent to the tumor. The AptaB4 and AptaB5 aptamers recognized the highest number of invasive carcinoma samples and metastatic carcinoma samples, respectively. Tissue adjacent to the tumor is often used as a negative control in histological analyses, but it is important to note that, despite presenting morphological characteristics of healthy tissue, transcriptome studies of adjacent tissue samples reveal increased expression of some genes that activate signaling pathways common to cancer. Thus, they can be considered intermediaries between healthy tissue and tumor tissue [32,33]. The identification of tumor-specific aptamers is outstanding and with great urgency, especially when detecting metastases. In addition, aptamers are also potential tools for drug delivery, dramatically reducing the side effects of nonspecific tumor therapy. 

After selecting aptamers with specificity and affinity for BC tumor cells, it is important to understand how the aptamer interacts with its targets. In this scenario, in silico methodologies are gaining more and more prominence, as they allow for the analysis of the recognition and binding stability of aptamer–target [34,35]. The aptamer interaction with its target depends entirely on its three-dimensional (3D) structure. We therefore used computational approaches to predict both the two-dimensional (2D) and 3D structures of aptamers. To do this, we relied on two servers: mFold and NUPACK. When comparing the results obtained from these two servers, we noticed a notable discrepancy in the structural predictions, except in the case of AptaB2. 

As a result, we chose to select the aptamers with the lowest free energy, as determined using the mFold server. To increase the reliability of the conformation adopted by the aptamer, we subjected the predicted structures to a molecular dynamics process, in which it was observed that AptaB5 adopted two predominant stable conformations. At the end of the molecular dynamics process, we were able to obtain the most likely conformation for each aptamer. Although aptamers can bind to a variety of molecules, in our study, we focused on membrane protein targets as we used the cell-SELEX method to select the aptamers. We therefore focused our efforts on identifying overexpressed proteins present in the MDA-MB-231 cell line. After using the pipeline filters to confirm the cellular localization and the expression levels in the other lineages recognized by the aptamers, four possible targets were proposed followed by the molecular docking assays. 

To define the best complex for each aptamer, we used the Haddock score as a criterion. This score is the sum of the Van der Waals, electrostatic, desolvation and restriction violation energies, and the surface area that interacts with the ligand. Thus, the value represents the overall analysis of the binding force [36]. The molecular docking methodology was used by Niazi et al. to identify the best aptamer and HER2 protein binding complexes, and the average Haddock score of the best complexes ranged from −57 to −47 [37]. Here, the proteins identified as potential targets for aptamers have been previously associated with cancer. CSKP, a peripheral plasma membrane protein, showed strong interaction with AptaB4; we obtained remarkable in silico results with the CSKP-AptaB4 complex, indicating that CSKP is indeed the AptaB4 target. Thus, it would be important to further validate these data using bench approaches, such as surface plasmon resonance and calorimetry methods. CSKP is described as participating in the regulation of cell proliferation and remodeling of the cytoskeleton. Increased CSKP expression is associated with a more aggressive phenotype and unfavorable clinical outcomes in colorectal cancer. In vitro silencing of CSKP in pancreatic tumor cells inhibits cell proliferation. In vivo experiments carried out with CSKP knockout revealed the activation of apoptosis pathways. CSKP is therefore a prospective biomarker with prognostic significance and a candidate for targeted therapy in liver cancer [38,39,40]. Another interesting result was observed with the TM9SF3 and AptaB5 complexes. However, the Haddock score values revealed that the conformation adopted in Cluster 2 (AptaB5.2) scored better than that adopted in Cluster 1 (AptaB5.1). This finding reveals the importance of the conformation adopted by the aptamer for interacting with the target molecule. Transmembrane protein 9, a member of family 3, has been reported to be overexpressed in several tumor types. Analysis of the expression of this protein in clinical specimens showed high expression of TM9SF3 in triple-negative breast tumors when compared with the expression observed in tissues adjacent to the tumor. Functional analysis results show that protein depletion inhibits the proliferation and migration of triple-negative lineage cells [41]. In addition, the in silico approach is increasingly being used in aptamer research. The computational biology field can make great strides related to aptamer technology, from the selection step to the optimization for better interaction with its specific target. As a result, it is possible to make the necessary modifications to improve the aptamer function. With these results, we hope to contribute to the ratification of aptamers as biosensors to be used in laboratory tests for breast cancer screening.

## 4. Materials and Methods

### 4.1. Cell Lines and Culture Conditions

Breast tumor cell lines MDA-MB-231, MDA- MB-468, HCC-70, and HCC-1937 were cultured with RPMI 1640 supplemented with 10% fetal bovine serum and 100 Ul/mL penicillin/streptomycin (Sigma, San Antonio, TX, USA—P4333); BT-474, MCF-7, and HCC-1954 cells were cultured with DEMEM supplemented with 10% fetal bovine serum and 100 Ul/mL penicillin/streptomycin (Sigma, TX, USA—P4333); MCF-10A was cultured with MEGM™ Mammary Epithelial Cell Growth Medium (Lonza/Clonectics Corporation, Basel, Switzerland—CC-3150—MEGM BulletKit), + BPE; rhEGF; hydrocortisone; insulin; penicillin/streptomycin at 100 Ul/mL (Sigma, TX, USA—P4333) and cholera toxin at 10 ng/mL (Sigma, TX, USA—C8052).

### 4.2. Cell-SELEX

Triple-negative breast tumor cell lines (MDA-MB-231) and their respective non-tumor control cells (MCF-10A) were used (Figure 15). Aptamers specific to the MDA-MB-231 tumor cell lines were previously selected by our group using the Cell-SELEX method after 12 rounds of selection [42]. Successive rounds of Cell-SELEX resulted in the selection of aptamers specifically recognized by breast tumor cells. A library of N30 oligonucleotide DNA aptamers (5 nmol) was initially incubated with the tumor cells for screening and selection of aptamers that bind to molecules present on the cell surface. The bound aptamers were then amplified in a PCR reaction. After 12 rounds of selection and amplification, we obtained the exponential enrichment of specific aptamers that were submitted to sequencing.

### 4.3. Identification and Analysis of Selected Aptamer Sequences for MDA-MB-231 Cells

The identification of aptamer sequences for breast tumor cells was generated using next-generation sequencing (NGS) methodologies, MiSeq (Illumina, San Diego, CA, USA) on the NGS Plataform of the Rede de Plataformas Tecnológicas Fiocruz, at the Laboratório de Genômica Aplicada e Bioinovações (IOC/Fiocruz). The quality of generated sequences was analyzed for the last five rounds (R8, R9, R10, R11, and R12) using the FastQC tool [43]. The results of the FastQC analyses were visualized in HTML format files containing sequences’ basic statistics, such as the number and size of generated reads, the distribution of quality values for each one of the bases, and the GC content. Nextera XT adapters, used to anchor the sequencing primers and low-quality sequences were removed using Trimmomatic software (Version: 0.39) [43]. Among the options available for running the Trimmomatic was SLIDINGWINDOW with the 4:20 option so that the end sequences were cut whenever the average quality was less than 30 (Phred Quality Score, Q ≥ 30) in four base intervals. The option MINLEN 74 was also used so that the sequences were eliminated if, after filtering, they had a length less than 74 bases, corresponding to the minimum expected size of the sequence. After trimming, the sequences were re-evaluated with FastQC to verify the filtering efficiency. Sequences trimmed in fastq format were converted to fasta format with the seqtk tool [44] to evaluate frequencies in each run. Thus, it was possible to analyze the generated data with shell script and obtain a manageable and organized database of candidate aptamer sequences.

### 4.4. Evaluation of Specificity of Aptamers Using Flow Cytometry

After identifying the sequences of the selected aptamers specific for MDA-MB-231 cells, the five most abundant sequences present in the last round of selection were evaluated for their specificity. MDA-MB-231 and MCF-10A cells were dissociated in Trypsin/EDTA (0.25%, Sigma, TX, USA) and counted in an automated cell counter (Countess II, Thermo Fisher Scientific, Waltham, MA, USA), and then 5 × 10^5^ cells were incubated with FAM-conjugated aptamers at concentrations of 25, 50, 100, 200, and 400 nM for 1 h in 5% CO_2_ at 37 °C. Binding buffer solution (DPBS, 4.5 g glucose, 1 g BSA, and 5 mL 1 M MgCl_2_) and the initial library with random sequences at concentration 400 nM were used as a negative control. The samples were analyzed on a FACScanto flow cytometer at the Flow Cytometry platform of Fiocruz—Rede de Plataformas Tecnológicas Fiocruz, with FlowJo software version 10. Three independent replicates were carried out for each test.

### 4.5. Analysis of the Dissociation Constant (Kd)

In order to analyze the binding affinity of the aptamers within tumoral and non-tumoral cells, the Kd value of total binding was verified using the median fluorescence intensity (MFI) obtained using flow cytometry. The analysis was performed using Prisma 5.0 and applying the equation (Bmax × X/(X + Kd) + NS × X + Background), where Bmax is the maximum number of binding sites, NS is the value of nonspecific binding, and X is the concentration of the aptamers used in the assay. The nonspecific binding was calculated from control samples incubated with no aptamers or with the initial aptamer library in 400 nM. Three independent replicates were carried out for each test.

### 4.6. Aptafluorescence

Breast tumor cell lines MDA-MB-231, MDA-MD-468, MCF-7, HCC-70, BT-474, HCC-1954, and HCC-1937 and the non-tumor cell line MCF-10A were plated (5 × 10^4^) in Labtek chamber slides. They were subsequently blocked with binding buffer solution (DPBS, 4.5 g glucose, 1 g BSA, and 5 mL 1 M MgCl_2_) and incubated with FAM-conjugated aptamers at 400 nM for 1 h in 5% CO_2_ at 37 °C. After the incubation time, the cells were washed with PBS and fixed in 4% paraformaldehyde for 5 min. The cells were incubated for 10 min with DAPI (1:5000) to counterstain nuclei, washed again in PBS, and mounted with DABCO. Image acquisition was performed on a Nikon Eclipse microscope (Tokyo, Japan) with Nikon Instruments software (Software Version 7.7.4). To analyze aptamer localization, 300 cells were observed. Three independent replicates were carried out for each test.

### 4.7. Three-Dimensional In Vitro Tumor Model

The 96-well configuration from n3D Biosciences, Inc. (Houston, TX, USA) was used to construct the 3D in vitro breast tumor model. The breast tumor cell line MDA-MB-231 and the non-tumor cell MCF-10A were plated (1 × 10^6^) in a 6-well plate and incubated with magnetic nanospheres at a ratio of 76 μL of nanosphere for every 1 × 10^6^ cells. After 24 h, the cells were then detached and plated (3 × 10^4^ cells per well) in a 96-well non-adherent plate. Next, the plate was placed over a magnet plate, and the spheroids were formed after 24 h. After this time, the spheroids were fixed with 4% paraformaldehyde under gentle stirring. For the binding assay, the spheroids were incubated for 2 h with the FAM-conjugated aptamers at 400 nM and with the aptamer initial library, diluted in binding buffer solution, under gentle shaking at room temperature. Next, the spheroids were washed with PBS and incubated with DAPI (Thermo Fisher Scientific, Waltham, MA, USA) for 20 min (1:5000) under gentle agitation, and then washed and finally incubated with DABCO (Sigma, TX, USA). Images were acquired by the High Content Screening System imagexpress Micro Confocal equipment (Thermo Fisher Scientific, Waltham, MA, USA) of the Bioassays platform of Fiocruz-Rede de Plataformas tecnológicas, Fiocruz, Brazil. Two independent replicates were carried out for each test.

### 4.8. Validation of Aptamer Recognition in Breast Cancer Samples Using Tissue Microarray (TMA)

The assays using clinical specimens from patients with breast cancer were performed with tissue microarray slides purchased from US Biomax Inc. (Rockville, MD, USA) (ref.: BR1008b). The slides were kept at 60 °C for 2 h before use. For the deparaffinization step, the slides were bathed in xylol (3× for 5 min), and then the sections were dehydrated in decreasing concentrations of ethanol (100%, 95%, 80%, and 70% for 5 min each). Then, the tissues were hydrated in water for 5 min. For antigenic recovery, the slides were placed in a streamer containing Citrate pH 6.0/Tris EDTA pH 9.0/Trilogy™ (Cell Marque, TX, USA) solution for 30 min. Then, after washes with TBS (3× for 5 min), the nonspecific binding was blocked with Novolink™ Protein Block (Leica Biosystems, Sao Paulo, Brazil) for 5 min. Then, the slides were incubated with FAM-conjugated aptamers AptaB1, AptaB2, AptaB3, AptaB4, and AptaB5, at a concentration of 400 nM for 1 h at room temperature and then washed with TBS and incubated with DAPI (1:5000 for 10 min). After washing in TBS, ProLong™ Gold Antifade (Thermo Fischer, Waltham, MA, USA) was used to mount the slides. After 24 h of polymerization, an analysis was performed in a Nikon Eclipse microscope (Tokyo, Japan), and the images were obtained using Nikon Instruments software (Software Version 7.7.4). 

### 4.9. Analysis of the Efficiency of Recognition of Aptamers for Diagnostic Application

To evaluate the accuracy and efficacy of aptamers as a diagnostic tool, we calculated the diagnostic index to identify the predictive value of aptamers and the best combination between them for application in a diagnostic test. Thus, the number of samples recognized by aptamers were quantified such as sensitivity: TP/(TP + FN), specificity: VN/(FP + TN), and accuracy: (VP + TN)/(TP + FP + FN + TN), where tumor samples with aptamer labeling were considered true positive (TP), tumor samples without aptamer labeling were considered false negative (FN), non-tumor samples with aptamer labeling were considered false positive (FP), and non-tumor samples without aptamer labeling were considered true negative (TN). 

### 4.10. Three-Dimensional Structural Characterization of Aptamers

To predict the 2D structures of the aptamers, we input the sequences into the UNAfold Web Server using the mFold extension (http://mFold.rna.albany.edu/?=mFold, accessed on 10 December 2022) available on the server’s website. Subsequently, to obtain a second prediction and compare the results, we utilized the NUPACK server (http://www.nupack.org, accessed on 12 December 2022). Both servers provided predictions of the secondary structure of aptamers in the parenthesis–dot code format, along with corresponding free energy values denoted as ΔG. 

Due to the limited availability of methodologies for predicting the 3D structural attributes of ssDNA aptamers in the existing literature, a critical transformation step was implemented. This involved converting the nucleotide sequences from DNA to RNA and subsequently switching back to DNA. The 3D structural predictions of aptamers followed a methodology adapted from Jeddi and Saiz (2017) [45]. The initially derived secondary structures in parenthesis–dot code format were used and submitted, along with the respective nucleotide sequences, to the RNA composer server (https://rnacomposer.cs.put.poznan.pl/, accessed on 20 December 2022). Thymine bases were replaced with uracil during this conversion. The outcome of this step produced the tertiary structure of RNA aptamers.

To obtain the original DNA sequence and subsequent 3D structure of the DNA aptamer, the x3DNA server (http://web.x3dna.org/index.php/mutatio, accessed on 21 December 2022) was utilized. The server assisted in converting uracil bases back into thymine and ribose sugars back into deoxyribose. Following this, the pdb file representing the 3D structure underwent refinement for geometry minimization using Phenix software, version 1.19.2-4158 (2-25-2021) [46]. 

### 4.11. Molecular Dynamics Simulations of the Aptamers

Molecular dynamics (MD) simulations were performed using Amber 22 [47,48] software with the FF14SB forcefield, adding the parameters of parmbsc1 force field for DNA simulation. Molecular topologies of the complexes were created using the tLEaP tool [47,48].

The systems were placed in triclinic boxes filled with TIP3P water molecules. The minimum distance from the solute to the edge of the box was set to 12 Å. Sodium and chloride ions were added to neutralize the systems. Electrostatic interactions were treated using the Particle Mesh Ewald (PME) method [49,50], with a cutoff of 10 Å, while the switching approach was applied to treat interactions between unbound atoms.

We executed cycles of geometry optimization and energy minimization, using the steepest descent algorithm for the initial 1000 steps of each cycle, followed by the conjugate gradient method for the remaining steps. The maximum number of steps was set at 5000 as a stop condition. The positions of the heavy atoms in the aptamer and the ligand were initially kept restricted using a harmonic potential of the force constant of 10 kcal/(mol, and we gradually decreased the force constant in each cycle until we achieved an unrestrained state of the molecules.

We raised the systems’ temperature from 20 to 300 K, with a cap of 500,000 steps during the heating simulation procedure. The weak coupling approach [51,52] was used to maintain the pressure at 1 atm. Ions and water molecules were kept free while the positions of the heavy atoms were constrained using a harmonic potential with a force constant of 10 kcal/(mol). Initial velocities at T = 20 K were obtained using the Maxwell–Boltzmann distribution. We limited this procedure to 500 picoseconds and used an integration time of two femtoseconds.

After the heating process, 4 ns long equilibrium dynamics were conducted. To control the position restraints of the heavy atoms, we used force constants. The force constants started at 5 kcal/ (mol), and we decreased them toward zero over the equilibrium phase. A Langevin thermostat was used to maintain the temperature at 300 K with a collision frequency of 2.0 ps.

A molecular dynamics simulation was performed using the NPT statistical ensemble. A Berendsen barostat [53,54] with an isotropic position scale was used to regulate the pressure. We generated three replicates with 500 ns long in each of them.

### 4.12. Trajectory Analysis

Root mean square deviation (RMSD), root mean square fluctuation (RMSF), and the clustering analysis were performed with Gromacs [52,55]. The clustering analysis was performed using the GROMOS method with a cutoff of 2.5 Å.

### 4.13. Selection of Candidate Proteins for Possible Targets Recognized by the Aptamer Using an In Silico Approach

To identify potential aptamer targets, we utilized Lawrence et al.’s (2015) entire proteomic dataset to identify proteins that exhibited double the expression in the tumor lineage (MDA-MB-231) compared with the non-tumor lineage (MCF-10A) [16]. In order to validate the overexpression of the selected proteins and select proteins that are attached to the cell membrane, we utilized proteomic data obtained from Ziegler et al. (2015) [17]. In addition, we predicted the cellular localization of the proteins using Psort II predictor and Uniprot (https://www.uniprot.org/, accessed on 10 December 2022). Following these steps, we identified 40 membrane proteins overexpressed in MDA-MB-231 compared with the MCF-10A cell line. A third filtering step was applied based on the cellular recognition by the five individual aptamers using the data previously published by Lawrence and colleagues (2015), and finally, 4 proteins were selected.

### 4.14. Obtaining the Three-Dimensional Structure of the Candidate Proteins for Molecular Docking

Molecular docking was carried out to predict the possible binding of the aptamer with the four selected proteins, and thus the three-dimensional structure of the target protein had to meet specific criteria. Therefore, we obtained the structure of the selected proteins in the Protein Data Bank (PDB—https://www.rcsb.org/, accessed on 12 December 2022) and AlphaFold (https://alphafold.ebi.ac.uk, accessed on 15 December 2022) [56]. Only structures with a prediction quality index defined by AlphaFold as very reliable or reliable (pLDDT > 70) progressed to the molecular docking phase.

### 4.15. Spatial Orientation Analysis and Electrostatic Potential Characterization of Selected Proteins

The Psort II server (https://psort.hgc, accessed on 22 December 2022) was utilized to verify the spatial orientation of plasma membrane proteins, with the membrane topology identified using the MTOP algorithm. To determine the protein’s spatial orientation, we used the PPM 2.0 web server, accessible on the OPM database (https://opm.phar.umich.edu/, accessed on 26 December 2022). An electrostatic potential analysis was carried out using the APBS program (https://server.poissonboltzmann.org, accessed on 23 December 2022) [57]. The PDB2PQR server, available in APBS, was used to add the charge and radius parameters of the Amber Force Field. The electrostatic potential of the protein was represented and colored on a scale from −10 (red) to +10 (blue) using the ChimeraX program, version 1.6.1. 

### 4.16. Molecular Docking

The protonation state of the selected proteins was predicted using the H++ web server (http://newbiophysics.cs.vt.edu/H++/index.php, accessed on 28 December 2022). In addition, the GETAREA server (https://curie.utmb.edu/getarea.html, accessed on 28 December 2022) [58] was used to identify residues with relative solvent accessibility >80%, thus identifying the active residues of the protein. For aptamers, the active residue was defined as the entire aptamer (76 nucleotides). The HADDOCK 2.4 server was used to run and analyze the molecular docking in standard mode. 

### 4.17. Molecular Dynamics Simulations of the Complexes

The complexes obtained after the molecular docking step were subjected to molecular dynamics simulations using the structures obtained in the molecular docking as initial conformations. Charmm-gui’s Membrane Builder tool [59,60] was used to construct the systems and insert a double-layer membrane of 1-palmitoyl-2-oleoyl-sn-glycero-3-phosphocholine (POPC) into each of them. Then, the structures generated with charmm-gui were used as input for the tleap program with the addition of the parameters of the lipid21 force field for the simulation of lipids. The remaining stages of the preparation and execution of the molecular dynamics for the complexes were conducted using the same parameters described above for the aptamers. Then, we performed molecular dynamics simulations using the best structures obtained from the molecular docking step as initial conformations. 

### 4.18. Target–Aptamers Binding Evaluation

The MM/GBSA was performed in Amber software using 50 ns of each simulation for the calculations, using the optimized generalized Born (GB) model called the OBC model defined by igb = 2 [61,62,63,64]. For the energy decomposition, we applied a per-residue basis scheme, defined by idecomp = 2. We applied the MM/GBSA method to calculate the Gibbs free energy change ΔG. ΔG was obtained according to the following equation: ΔG=ΔEvdw+ΔEele+ΔGesurf+ΔGegb.
where ΔEvdw stands for the Van der Waals interaction change, ΔEele is the electrostatic energy change, ΔGesurf accounts for the surface free energy change, and ΔGegb denotes the generalized Born free energy change.

## 5. Conclusions

In conclusion, the data presented in this study reinforce the capacity of aptamers as a promising diagnostic tool for breast cancer. The selected aptamers recognized all molecular subtypes of breast cancer and, moreover, the different intrinsic subtypes of the triple-negative breast tumor. The validation of the recognition in clinical samples at different stages confirms the possibility of using aptamers as a strategy for more accurate diagnostic and prognosis of the disease. In the near future, aptamers will open the avenue for the minimally invasive BC diagnosis based on liquid biopsy. Finally, the in silico methodology showed the potential to be used to determine possible aptamer targets, and CSKP could then be suggested as a new biomarker for BC.

## 6. Patents

The aptamer sequences are protected under patent filing PCT/BR2022/050356.

## Figures and Tables

**Figure 1 ijms-25-00840-f001:**
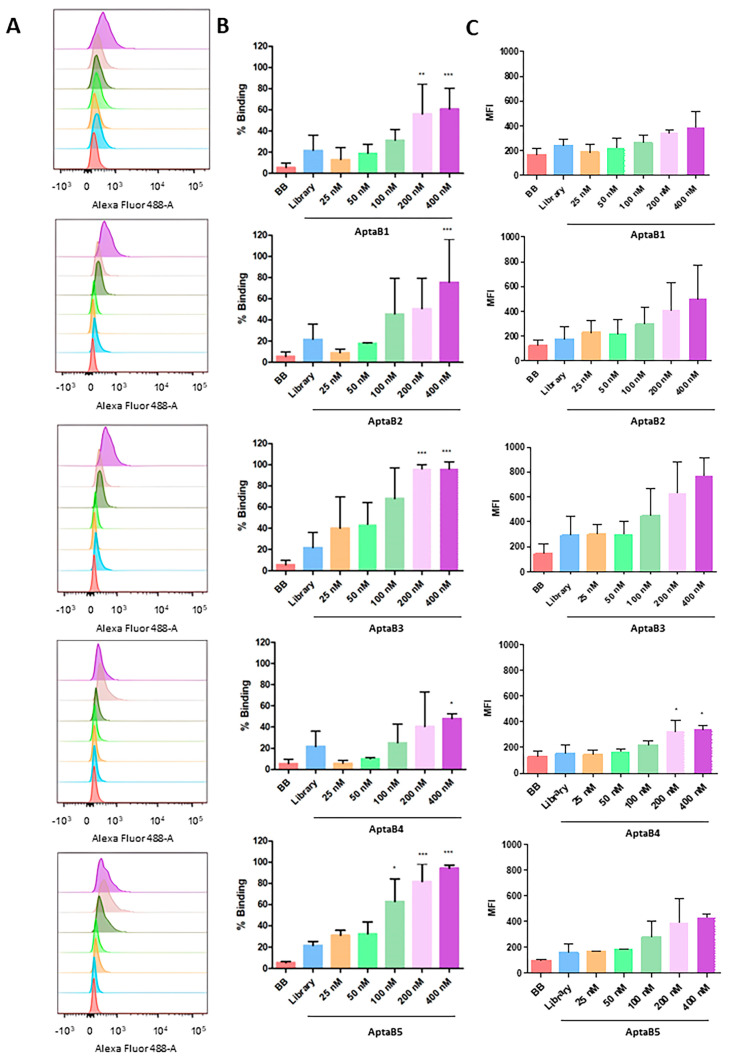
Analysis of the binding capacity of the aptamer panel for the MDA-MB-231 strain. Histogram illustrating the percentage of cells incubated with binding buffer solution, initial library, and increasing doses of AptaB1–AptaB5 (**A**). Graphical representation of the mean percentage of binding of AptaB1–AptaB5 aptamers (**B**). Graphical representation of the average MFI (**C**). * *p* < 0.05, ** *p* < 0.01, *** *p* < 0.001.

**Figure 2 ijms-25-00840-f002:**
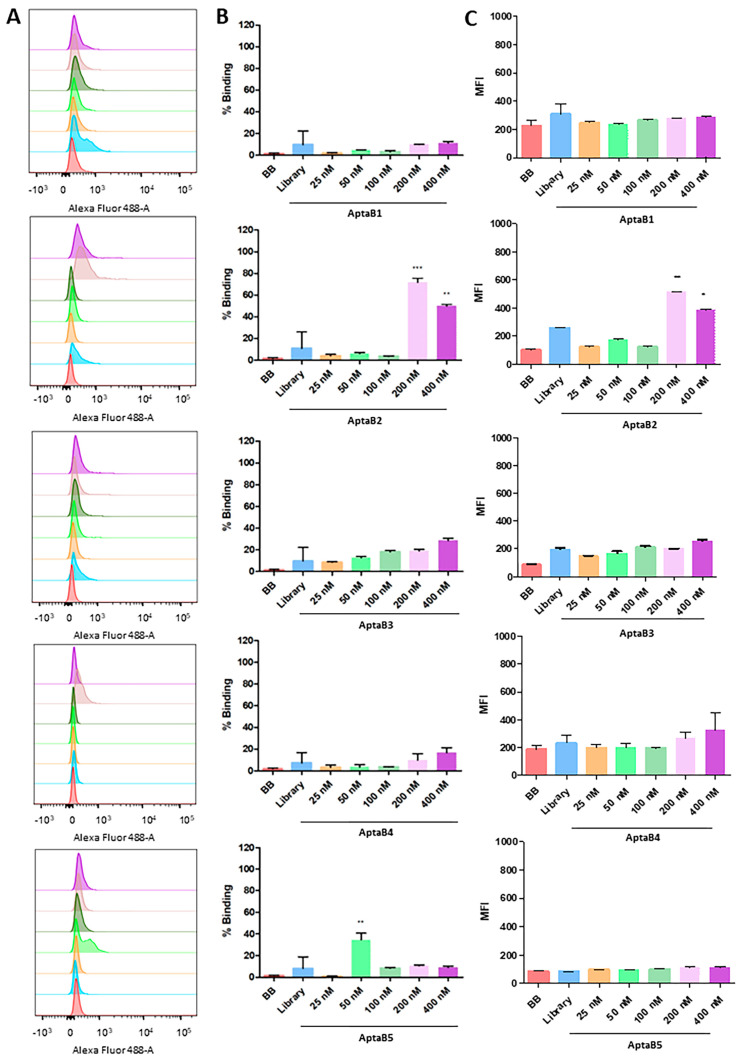
Analysis of the recognition specificity of the aptamer panel for the non-tumor line MCF-10A. Histogram illustrating the percentage of cells incubated with binding buffer solution, initial library, and increasing doses of AptaB1–AptaB5 (**A**). Graphical representation of the mean percentage of binding of AptaB1–AptaB5 aptamers (**B**). Graphical representation of the average of MFI AptaB1–AptaB5 (**C**). * *p* < 0.05, ** *p* < 0.01, *** *p* < 0.001.

**Figure 3 ijms-25-00840-f003:**
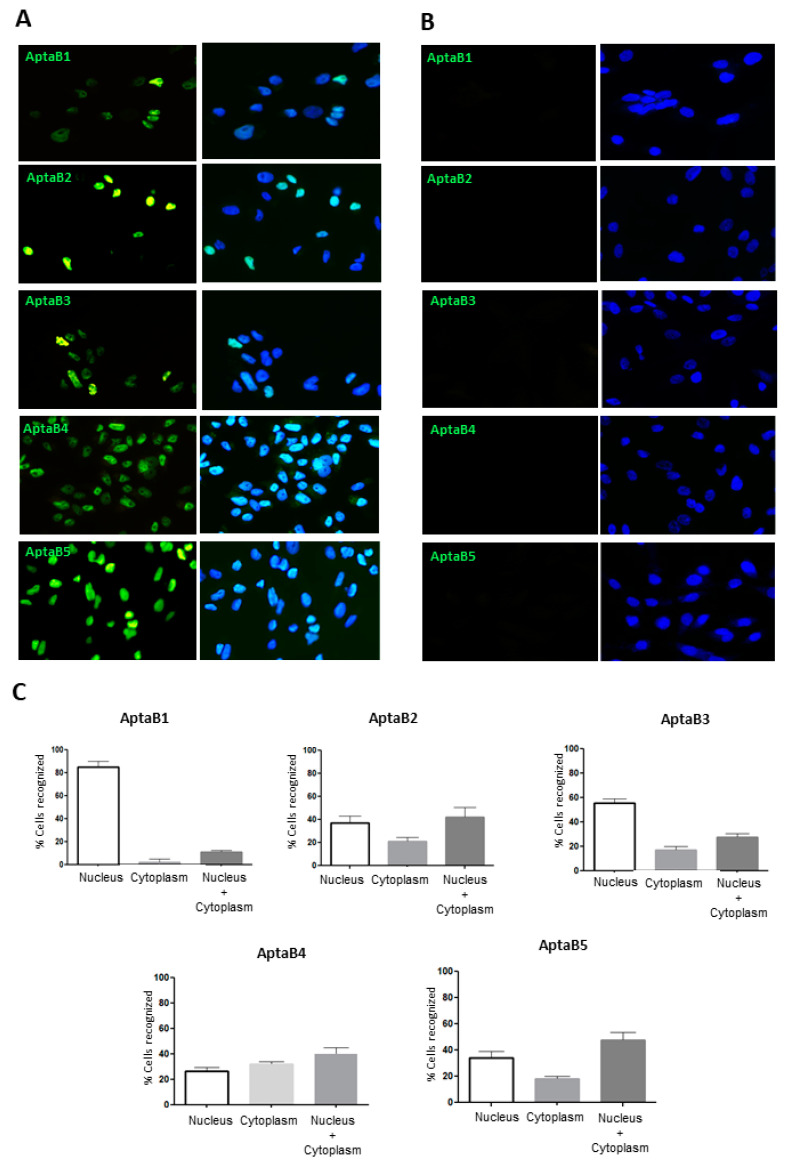
Validation of the recognition and specificity of the aptamer panel. AptaB1-AptaB5 aptafluorescence (green) in MDA-MB-231 (**A**) and the absence of labeling in MCF-10A (**B**); nucleus labeled with DAPI (blue). Analysis and quantification of the localization of labeling observed in MDA-MB-231 (**C**).

**Figure 4 ijms-25-00840-f004:**
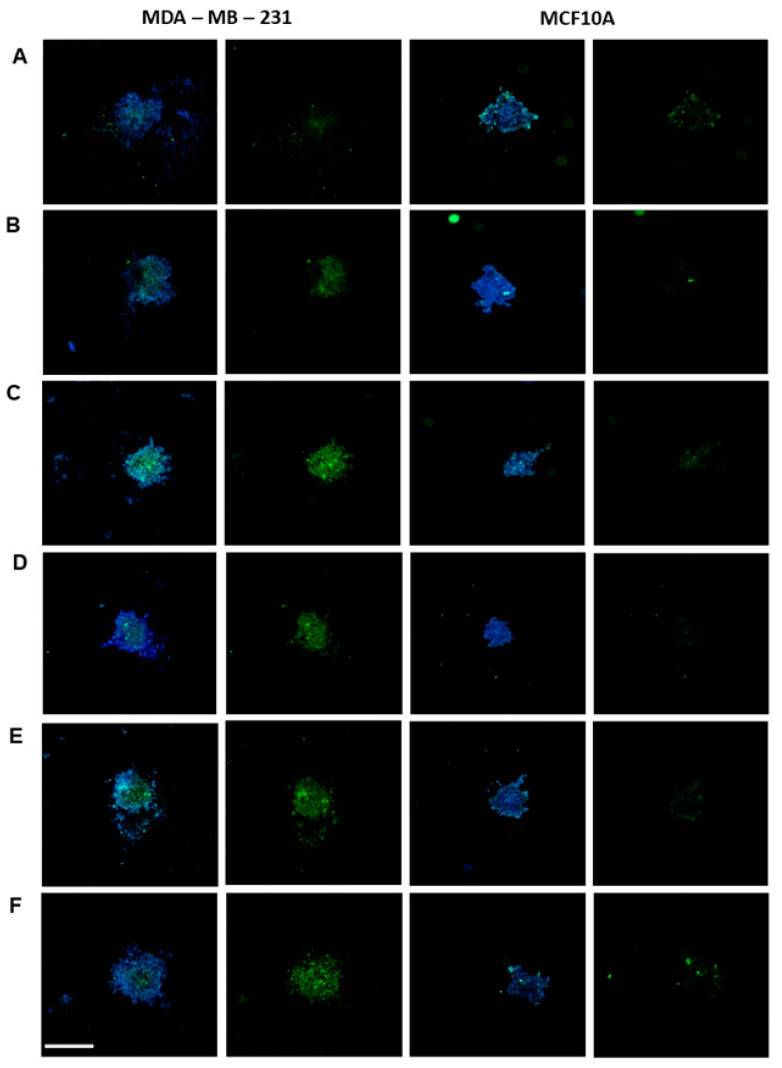
Aptafluorescence assay in a three-dimensional culture model. Core in blue (DAPI); aptamer conjugated to FAM in green. (**A**) Aptamer library, (**B**) AptaB1, (**C**) AptaB2, (**D**) AptaB3, (**E**) AptaB4, and (**F**) AptaB5. The images were obtained using imagexpress Micro Confocal equipment in a 10× objective. Bar = 200 µM.

**Figure 5 ijms-25-00840-f005:**
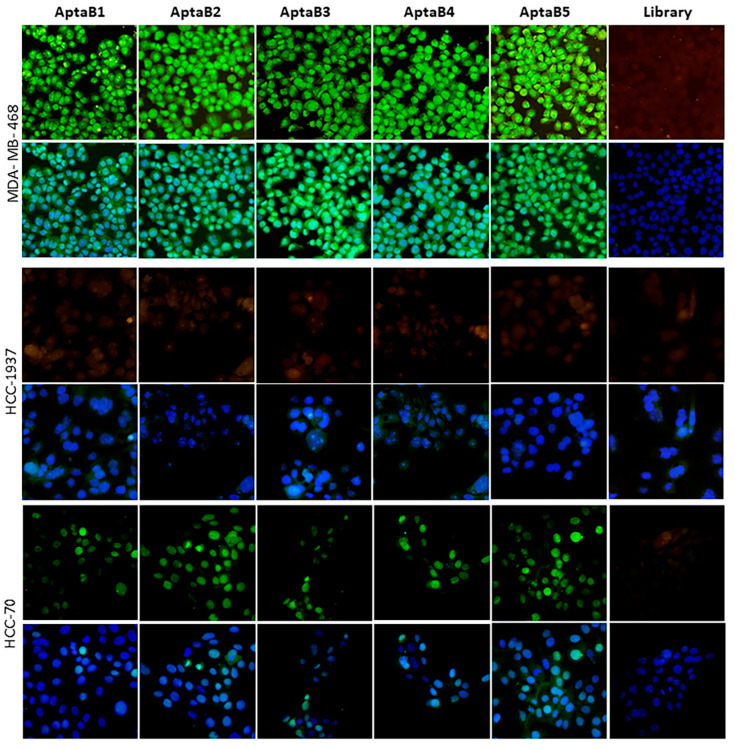
Evaluation of the detection capacity of aptamers in triple-negative cell lines. Aptafluorescence assay with the aptamers AptaB1–AptaB5 or with the starting library—FAM (green) using triple-negative cell lines MDA-MB-468, HCC-70, and HCC-1937; nucleus stained with DAPI (blue).

**Figure 6 ijms-25-00840-f006:**
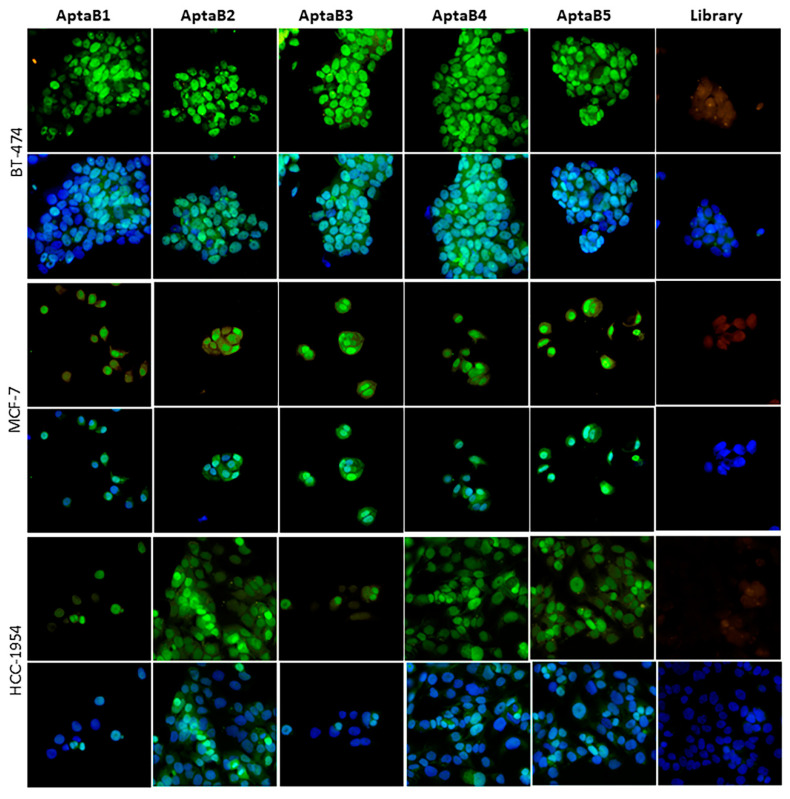
Evaluation of the detection capacity of aptamers in cell lines of the luminal A, luminal B, and HER2+ molecular subtypes. Aptafluorescence assay with the aptamers AptaB1–AptaB5 or with the starting library—FAM (green) using MCF-7 (luminal A), BT-474 (luminal B), and HCC-1954 (HER2) cell lines; nucleus stained with DAPI (blue). Representative images of the library—FAM recognition. Overlay of the library—FAM recognition image and DAPI nucleus stain.

**Figure 7 ijms-25-00840-f007:**
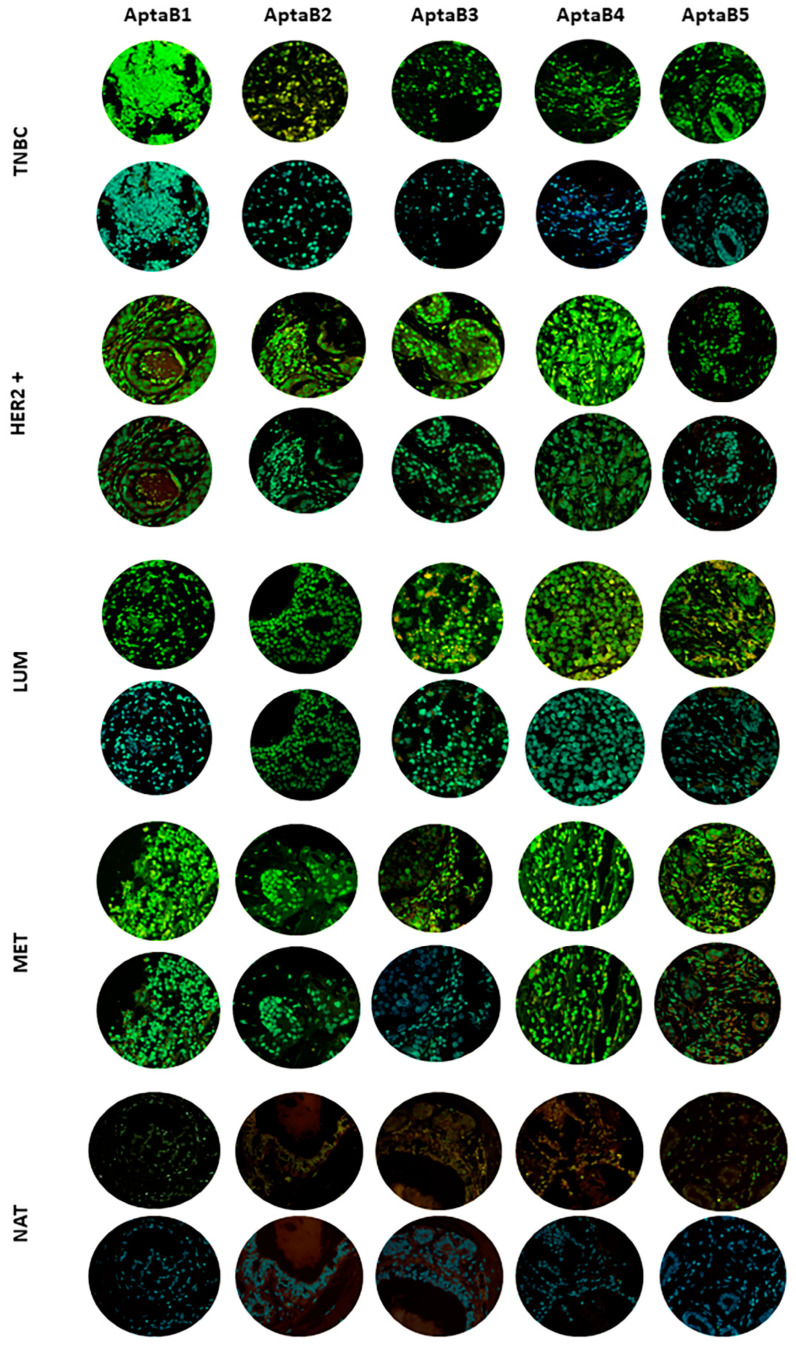
Validation of MD1–MD5 aptamer recognition in breast cancer samples using the tissue microarray technique (TMA). Triple-negative breast cancer (TNBC), HER2+ subtype (HER2+), luminal subtype (LUM), lymph node metastasis (MET), and adjacent normal tissue (NAT). Aptamers AptaB1–AptaB5 (green); cell nucleus stained with DAPI (blue).

**Figure 8 ijms-25-00840-f008:**
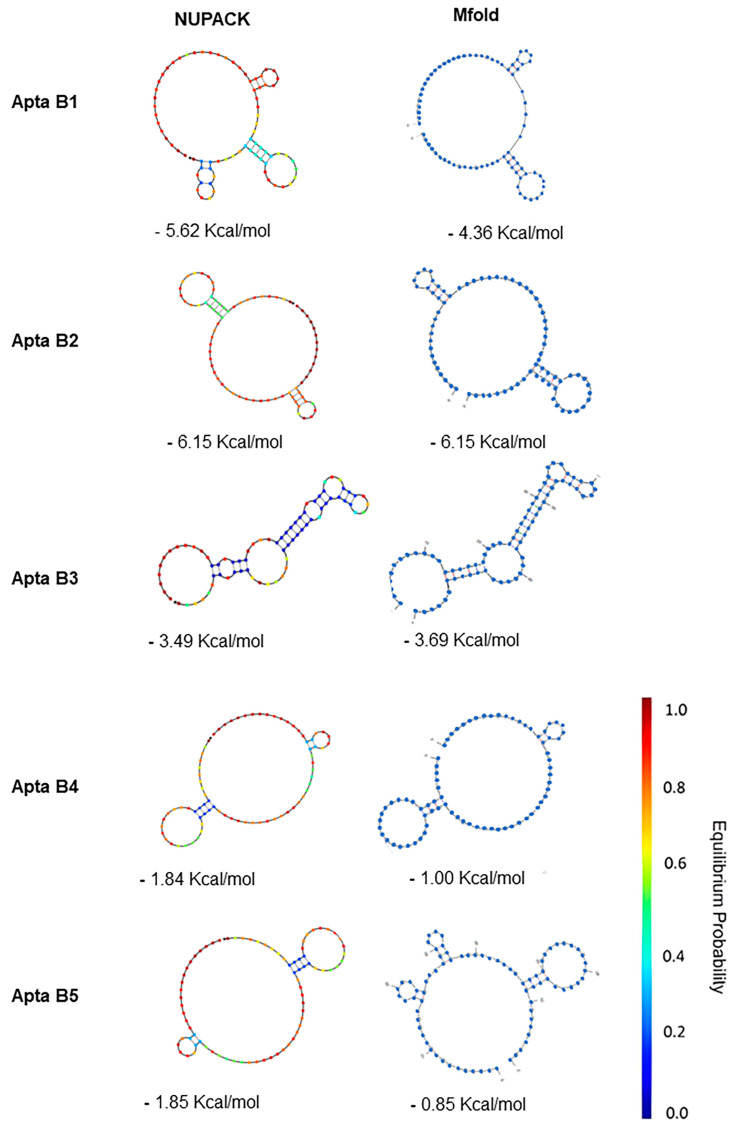
Representation of the secondary structure prediction of the aptamers selected for the MDA-MB-231 strain. The first column shows the results obtained with the NUPACK server and the second column shows the data from the mFold server. Below the structures, the free energy ΔG values are presented. For the structure obtained with NUPACK, the color of the dots is related to the equilibrium probability of paired bases, as showed in the equilibrium probability colored scale.

**Figure 9 ijms-25-00840-f009:**
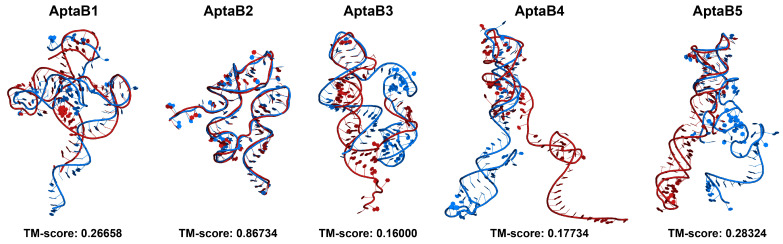
Comparison of the predicted tertiary structures of aptamers obtained with RNAcompose with 2D structures predicted using the mFold and NUPACK servers. Aptamer models generated from mFold 2D structures are shown in blue, while models generated from NUPACK are shown in red. Structural alignment and calculation of TM scores were performed using the RNA-align server to quantitatively assess the similarity between the 3D models from the two sources. Tertiary structure visualizations were generated using Pymol.

**Figure 10 ijms-25-00840-f010:**
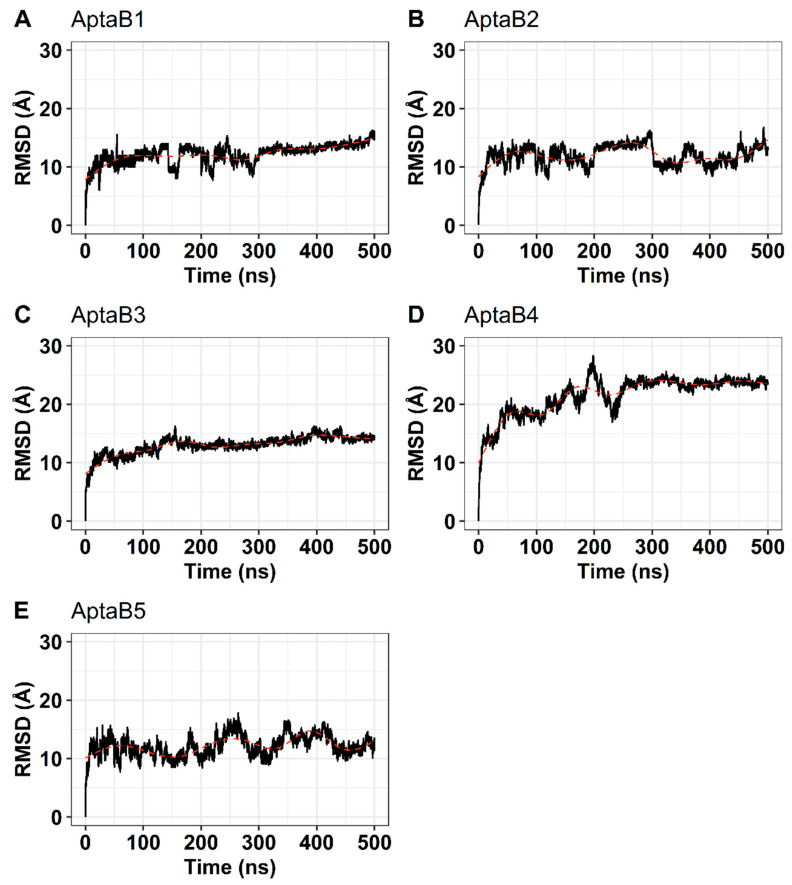
Analysis of conformational changes in structures during molecular dynamics simulation. The root mean square deviation (RMSD) is shown in Angstrons at the left, while the simulation time is given at the bottom. Black lines indicate the RMSD, while the shaded red line shows the tendency of the graph. (**A**): the RMSD of APTAB1 oscillated between 6 and 16 Å. The greatest variations happened between 30 ns and 300 ns. After 300 ns the RMSD stabilized around 13 Å and rose steadily until 16 Å. (**B**): The RMSD of AptaB2 did not stabilize throughout the simulation time. (**C**): AptaB3 was the less unstable aptamer among all 5. The RMSD of AptaB3 fluctuated around 14 Å and varied less than 4 Å for most of the simulation. (**D**): AptaB4 was quite unstable in the first half of the simulation, reaching 20 Å before the first 50 ns. Around 250 ns it stabilized near 24 Å and kept oscillating less than 4 Å until the end. (**E**): The RMSD of AptaB5 fluctuated around 13 Å, but showed great variations, reaching nearly 17 Å on its highest peak.

**Figure 11 ijms-25-00840-f011:**
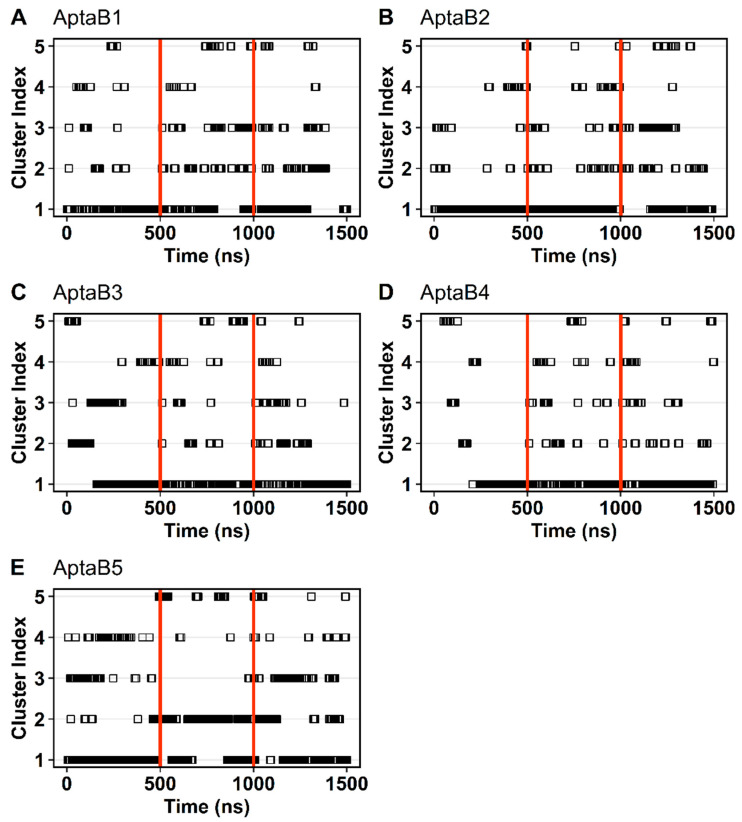
Clusters analysis based on the change in RMSD. Each cluster represents a group of conformations assumed along the molecular dynamic trajectory with AptaB1–AptaB5. The five biggest clusters are shown for each aptamer. Time is given at the bottom, while the cluster index is given at the left. Each structure is shown as a square. Black horizontal bars indicate a high density of squares grouped. Red vertical lines delimit the simulation time of the three replicates. (**A**): The five biggest clusters of AptaB1. The most expressive cluster comprises the three replicates indicating only one representative structure for the whole simulation. (**B**): Among the five biggest clusters of AptaB2, the cluster 1 also comprises the three replicates and appears as the most significant in all of them. (**C**): Cluster 1 in also the most expressive cluster in all three replicates of AptaB3. However, in the first 200 ns cluster 2 and cluster 3 are more frequent than cluster 1. (**D**): Among the five biggest clusters of AptaB4, cluster 1 is the most frequent cluster in all three replicates. However, in the first 250 ns, the conformations are clustered in several other cluster in despite of cluster 1. (**E**): The five biggest clusters of AptaB5. The cluster 1 is more frequent in the replicates 1 and 3, however, in the simulation time between 500 ns and 100 ns (corresponding to replicate 2) the cluster 2 is the biggest cluster formed. This result indicates the existence of two stable conformations for this aptamer.

**Figure 12 ijms-25-00840-f012:**
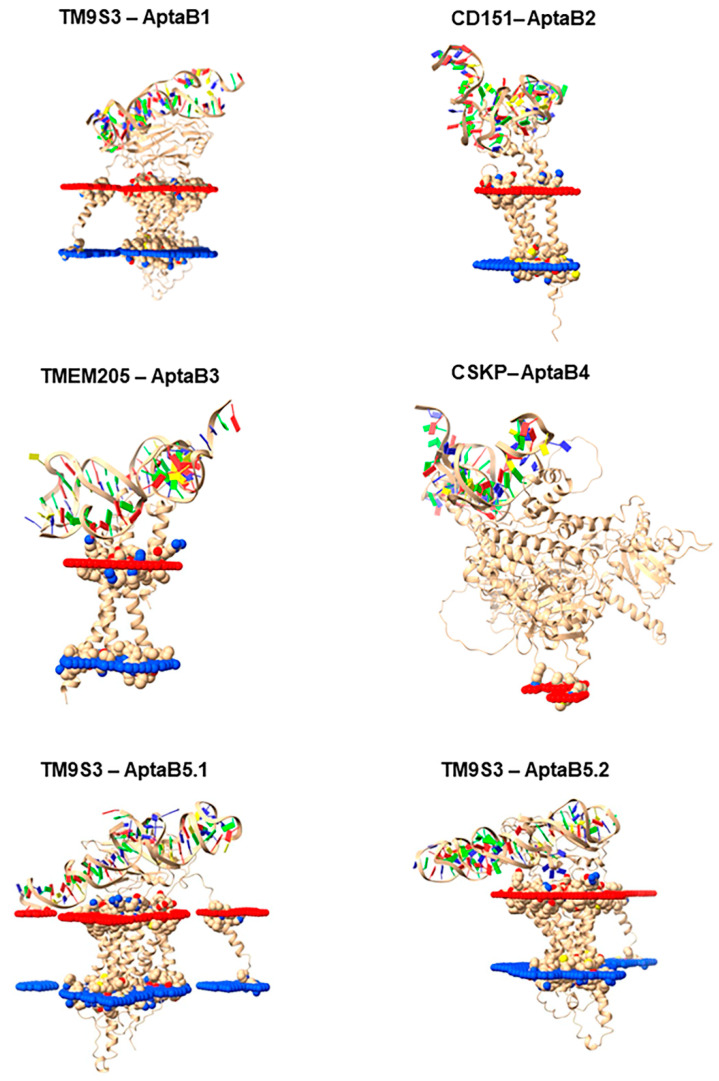
The best binding complexes for potential aptamer targets. Image representing the best clusters from the molecular docking step for the TMPS3–AptaB1, CD151–AptaB2, TMEM205–AptaB3, CSKPAptaB4, TM9S3–AptaB5.1, and TM9S3–AptaB5.2 complexes. For the spatial orientation, the red portion indicates the extracellular region and the blue portion indicates the intracellular region.

**Figure 13 ijms-25-00840-f013:**
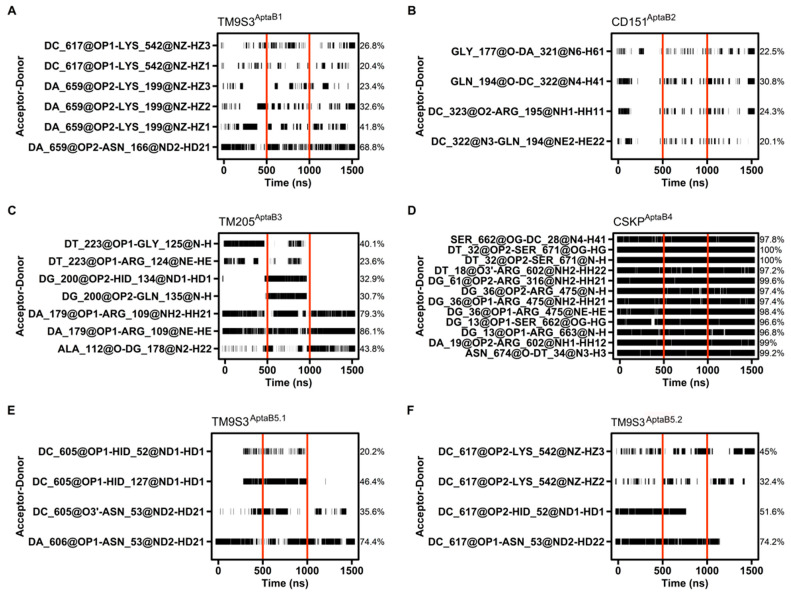
Hydrogen bond occupancy between the aptamer and the protein. The interacting pair of atoms are shown on the left, while the percentage of hbond occupancy is given on the right. The simulation time is shown in ns at the bottom. Horizontal black bars indicate each point of the simulation time that the interactions happened, whereas vertical red bars delimit each replicate time. Only hbond occupancies with 20% length or more are shown. (**A**): The binding of TM9S3 with AptaB1 is held mostly by the pair DA 659 and ASN 166. This interaction happens in all the replicates and covers 68.8% of the simulation time. 5 other weaker interactions above 20% hbond occupancy help to stabilize the binding throughout the simulation time. (**B**): In CD151^AptaB2^ there are just four interacting pairs with hbond occupancies over 20%, whereas the highest one happens for only 30.8% of the simulation. In several points of the simulation time there are no interaction happening at all. (**C**): In TM205^AptaB3^ There are 7 interacting pairs above 20% hbond occupancy. However, four of them does not happen in one of the replicates. (**D**): The system CSKP^AptaB4^ stands out in terms of hbond occupancy since its 12 higher hbond occupancy values are above 95%. (**E**): TM9S3^AptaB5.1^ formed 4 hbond with over 20% occupancy. Two of those hbonds does not happen in one of the replicates so the pair DA 606 and ASN 53 held the interaction for most of the time (74.4%). (**F**): similarly to TM9S3^AptaB1^, TM9S3^AptaB5.2^, also formed 4 hbond with over 20% occupancy and two of them also involved the same residues, ASN 53 and HID 52, being the former part of the most stable interacting pair (DC 617-ASN 53: 74.2%).

**Figure 14 ijms-25-00840-f014:**
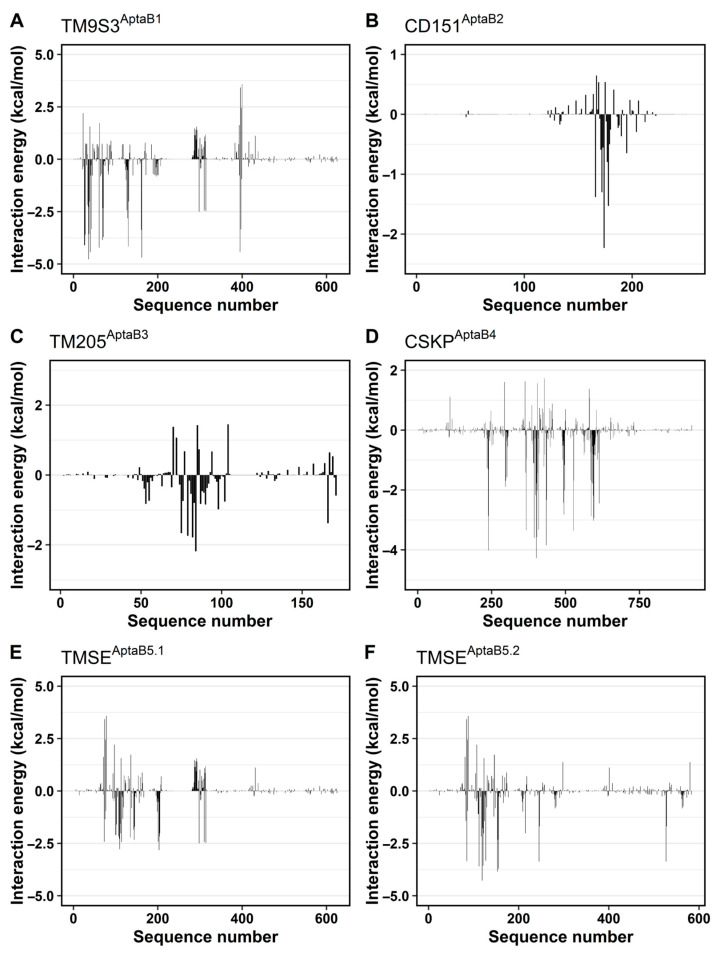
Decomposition of ΔG_bind_. The individual contribution of each residue to the total energy is depicted as vertical black bars. Negative energy values stand for residues working in favor of the binding process, while positive values denote residues disturbing the interaction. (**A**): Several residues are contributing to the binding of TM9S3 with AptaB1 reaching near 2.5 Kcal/mol. Contrastingly, highly positive energies are observed, particularly between residues 60 and 80. (**B**): Most of the residues in CD151 has nearly insignificant participation in the interaction with AptaB2. Despite showing some residues with highly negative energies, the number of residues repealing the ligand is superior, which resulted in the less negative ΔG_bind_ observed among all the systems. (**C**): In TM205^AptaB3^ the number of negative energies overcomes the positive ones. Besides the residues with positive energies do not reach 1.5 kcal/mol. (**D**): The system CSKP^AptaB4^ has considerably more residues with negative energies than residues with positive energies. Besides, several residues are reaching −3 kcal/mol, while a few one’s overpass −4 kcal/mol. In contrast, the most positive energies do not reach 2 kcal/mol resulting in a deeply negative ΔG_bind_. (**E**): TM9S3^AptaB5.1^ has some similarities with TM9S3^AptaB1^. However, the negative contributions and positive ones have lower absolute values. (**F**): TM9S3^AptaB5.2^ is like TM9S3^AptaB1^, but with higher absolute values. The residues with the most negative energies in TM9S3^AptaB5.2^ reached −5 kcal/mol almost while in TM9S3^AptaB5.1^ the most negative energies are close to −2.5 kcal/mol.

**Figure 15 ijms-25-00840-f015:**
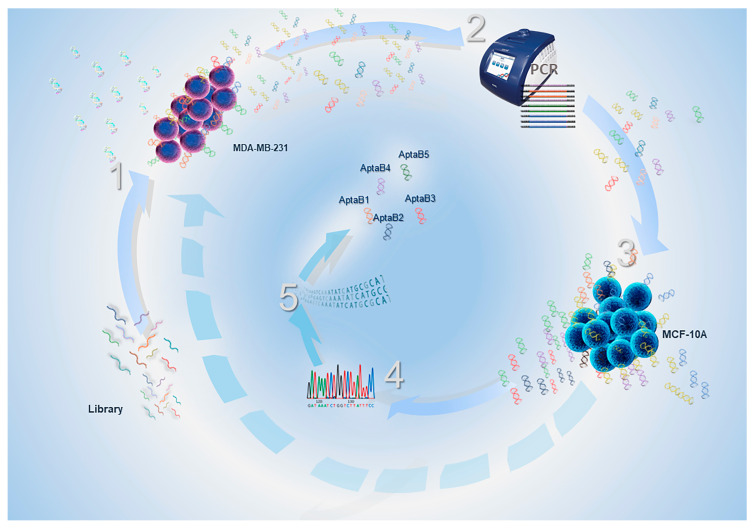
Summary schematic representation of Cell-SELEX (systematic evolution of ligands by exponential enrichment). 1—Initially, the DNA aptamer library (N30) was incubated with MDA-MB-231 target cells. 2—The cells were washed to remove unbounded sequences, and the bounded sequences were collected and amplified using PCR. 3—The aptamers went through the negative selection step in the MCF-10A cells, and the bound sequences were removed and eliminated, while the unbounded sequences were amplified using PCR and used in the subsequent Cell-SELEX rounds. 4—After 12 rounds of selection, the resulting aptamer pool was sequenced using the NGS methodology. 5—Finally, the aptamers were computationally analyzed, and the five most frequent sequences (AptaB1, AptaB2, AptaB3, AptaB4, and AptaB5) were selected for further validation steps.

**Table 1 ijms-25-00840-t001:** Dissociation constant (Kd) analysis of aptamer binding to the target cell MDA-MB-231.

APTAMERS	KD VALUES
APTAB1	139 ± 14 nM
APTAB2	206 ± 41 nM
APTAB3	145 ± 31 nM
APTAB4	194 ± 0.7 nM
APTAB5	126 ± 19 nM

**Table 2 ijms-25-00840-t002:** Recognition of tissue samples from primary tumor, metastatic tissue, and tissue adjacent to the tumor.

Aptamer	Sample	Recognition/ Total Number	% Recognition	Staining Intensity
AptaB1	Adjacent tissue	1/10	10%	(+)
	Primary tumor tissue	7/50	14%	(+++)
	Metastatic tissue	5/40	12.5%	(+++)
AptaB2	Adjacent tissue	1/10	10%	(+)
	Primary tumor tissue	18 /50	36%	(+++)
	Metastatic tissue	5/40	12.5%	(++)
AptaB3	Adjacent tissue	1/10	10%	(+)
	Primary tumor tissue	3/50	6%	(+++)
	Metastatic tissue	6/40	15%	(++)
AptaB4	Adjacent tissue	4/10	40%	(+)
	Primary tumor tissue	25/50	50%	(+++)
	Metastatic tissue	15/40	37.5%	(+++)
AptaB5	Adjacent tissue	3/10	30%	(+)
	Primary tumor tissue	20/50	40%	(+++)
	Metastatic tissue	27/40	67.5%	(+++)

(+) low labeling intensity; (++) moderate labeling intensity; (+++) high labeling intensity.

**Table 3 ijms-25-00840-t003:** Recognition of breast carcinoma primary sites according to molecular subtype.

Aptamer	Molecular Subtype from Primary Tumor	Recognition/ Total Number	% Recognition	Staining Intensity
AptaB1	Luminal	3/28	10.7	(+++)
	HER 2	3/7	42	(+++)
	Triple-negative	1/10	10	(+++)
AptaB2	Luminal	8/28	28	(+++)
	HER 2	5/7	70	(+++)
	Triple-negative	4/10	40	(+++)
AptaB3	Luminal	1/28	3.5	(+++)
	HER 2	1/7	14	(+++)
	Triple-negative	1/10	10	(++)
AptaB4	Luminal	12/28	42	(+++)
	HER 2	6/7	85	(+++)
	Triple-negative	4/10	40	(++)
AptaB5	Luminal	9/28	32	(+++)
	HER 2	5/7	70	(+++)
	Triple-negative	5/10	50	(+++)

(++) moderate labeling intensity; (+++) high labeling intensity.

**Table 4 ijms-25-00840-t004:** Recognition of metastatic lymph node tissue samples according to molecular subtype.

Aptamer	Molecular Subtype of the Metastatic Sample	Recognition/Number of Samples	% Recognition	Staining Intensity
AptaB1	Luminal	3/14	21%	(+++)
	HER 2	0/9	-	(−)
	Triple-negative	0/8	-	(−)
AptaB2	Luminal	1/14	7.1%	(+++)
	HER 2	2/9	22%	(+++)
	Triple-negative	2/8	25%	(+++)
AptaB3	Luminal	1/14	3.5%	(+++)
	HER 2	1/9	11%	(+++)
	Triple-negative	3/8	37.5%	(++)
AptaB4	Luminal	6/14	42%	(+++)
	HER 2	1/9	11%	(+++)
	Triple-negative	5/8	62%	(++)
AptaB5	Luminal	10/14	32%	(+++)
	HER 2	7/9	77%	(+++)
	Triple-negative	5/8	62%	(+++)

(−) absence of staining; (++) moderate labeling intensity; (+++) high labeling intensity.

**Table 5 ijms-25-00840-t005:** Recognition of aptamers according to the degree of tumor staging.

Stage	Number of Samples	AptaB1	AptaB2	AptaB3	AptaB4	AptaB5
I	4	0	1	0	3	2
II	39	4	15	3	20	4
III	3	2	1	1	2	3

**Table 6 ijms-25-00840-t006:** Recognition of aptamers according to the histological grade of the tumor.

Grade	Number of Samples	AptaB1	AptaB2	AptaB3	AptaB4	AptaB5
I	8	1	4	0	5	1
II	29	4	10	3	17	11
III	9	1	4	0	4	5

**Table 7 ijms-25-00840-t007:** Recognition of clinical samples by aptamers according to the TMN classification.

TNM	Number of Samples	AptaB1	AptaB2	AptaB3	AptaB4	AptaB5
T1N0M0	4	0	1	0	3	2
T2N0M0	29	4	11	2	15	10
T2N1M0	6	0	2	1	3	3
T2N3M0	1	1	0	1	0	1
T3N0M0	4	0	2	0	2	0
T3N1M0	1	0	0	0	1	1
T4N0M0	3	1	2	0	2	2
T4N1M0	2	1	1	0	1	2

**Table 8 ijms-25-00840-t008:** Diagnostic index calculated from the recognition of aptamers in primary, metastatic, and adjacent tumor tissue.

Aptamers	Sensitivity	Specificity	Accuracy
AptaB1	13%	90%	21%
AptaB2	26%	90%	32%
AptaB3	10%	90%	18%
AptaB4	44%	60%	46%
AptaB5	52%	70%	54%
AptaB4 + AptaB5	77%	40%	73%
AptaB2 + AptaB4 + AptaB5	87%	30%	81%
AptaB2 + AptaB3 + AptaB4 + AptaB5	89%	30%	83%
AptaB1 + AptaB2 + AptaB3 + AptaB4 + AptaB5	96%	30%	89%

**Table 9 ijms-25-00840-t009:** Haddock score for the result of the molecular docking with the selected proteins and representative structures of the AptaB1, AptaB2, AptaB3, AptaB4, AptaB5.1, and AptaB5.2 aptamers.

	Protein–AptaB1	Haddock Score
A	CSKP–AptaB1	−34.9 +/− 7.5
B	TM9S3–AptaB1	−76.7 +/− 33.4
C	TMEM205–AptaB1	−48.1 +/− 8.0
D	CD151–AptaB1	2.8 +/− 9.4
	Protein–AptaB2	Haddock Score
A	CSKP–AptaB2	−8.3 +/− 8.9
B	TM9S3–AptaB2	−11.1 +/− 18.0
C	TMEM205–AptaB2	−46.7 +/− 6.1
D	CD151–AptaB2	−53.1 +/− 9.2
	Protein–AptaB3	Haddock Score
A	CSKP–AptaB3	−8.7 +/− 21.7
B	TM9S3–AptaB3	−31.2 +/− 3.9
C	TMEM205–AptaB3	−34.2 +/− 3.5
D	CD151–AptaB3	17.3 +/− 9.9
	Protein–AptaB4	Haddock Score
A	CSK–AptaB4	−34.7 +/− 11.1
B	TM9S3–AptaB4	−22.3 +/− 4.0
C	TMEM205–AptaB4	−23.9 +/− 6.5
D	CD151–AptaB4	−8.4 +/− 25.8
	Protein–AptaB5.1	Haddock Score
A	CSKP–AptaB5.1	20.5 +/− 27.0
B	TM9S3–AptaB5.1	−41.2 +/− 16.2
C	TMEM205–AptaB5.1	12.3 +/− 11.5
D	CD151–AptaB5.1	18.0 +/− 13.2
	Protein–AptaB5.2	Haddock Score
A	CSKP–AptaB5.2	−4.7 +/− 22.0
B	TM9S3–AptaB5.2	−81.2 +/− 4.4
C	TMEM205–AptaB5.2	−3.7 +/− 13.9
D	CD151–AptaB5.2	6.5 +/− 6.5

**Table 10 ijms-25-00840-t010:** Binding free energy change (ΔG_bind_) for all complexes calculated using MM/GBSA.

System	ΔE_vdw_	Δ_ele_	Δ_egb_	ΔG_esurf_	ΔG_bind_
TM9S3^AptaB1^	−212.04	4821.73	−4654.34	−23.45	−68.1 ± 2.7
CD151^AptaB2^	−31.49	983.92	−952.11	−3.99	−03.6 ± 3.7
TM205^AptaB3^	−52.77	874.05	−842.43	−13.90	−35.0 ± 3.3
CSKP^AptaB4^	−246.40	5174.56	−5015.07	−30.90	−117.8 ± 2.2
TMS9^AptaB 5.1^	−209.31	4883.27	−4703.32	−24.24	−53.6 ± 2.1
TMS9^AptaB 5.2^	−210.58	4925.62	−4745.78	−28.66	−59.4 ± 2.5

## Data Availability

Data are contained within the article.

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
