# Peer review of "Aptamer-Based Recognition of Breast Tumor Cells: A New Era for Breast Cancer Diagnosis"

_ijms, 2024, doi:10.3390/ijms25020840_

Round 1

Reviewer 1 Report

Comments and Suggestions for Authors

The manuscript ijms-2674455 entitled “Aptamer-based recognition of breast tumor cells: a new era for breast cancer diagnosis” has identified 5 aptamers, which selectively recognice breast cancer cells, specifically MDA-MB-231 and they do not bind to non-tumoral cell. Aptamers validation studies showed that aptaB4 and aptaB5 have specific recognition for primary and metastatic tissues, respectively. Moreover, the recognition study was extended to other breast cancer subtypes and more complex culture models, such as 3D cultures. Finally, computacional biological studies were performed to identify the potential target protein of each one of the aptamers. The results indicate that the target of AptaB4 seems to be CSKP.

There are suggestions for majors corrections.

1-      The size of the letters in the legend of the graphic representation of the mean percentage in Figure 1 and 2 should be improved.

2-      On page 3, line 101, the concentration units (nM) are missing from the Kd value for AptaB2. Furthermore, the Kd values described on this same page and in figure 3 for AptaB2 (206 nM) and for AptaB5 (126 nM) do not match the values presented in the discussion, page 27, line 496.

3-      The replicates and number of assays carried out to obtain the binding values, the Kd and detection capacity of the five aptamers should be indicated.

4-      Taking into account the low recognition of non-tumor breast cells and the different pattern in the graph of the dissociation constants (Kd) of the aptamers compared with tumoral cells the values of the Kd shown are misleading These values of the constants must be removed.

5-      The sentence “all five apatamers recognized the luminal subtype with high intensity” needs to be more elaborated, considering that the recognition percentage for the Luminal subtype is 3.5% for the aptaB3 despite the fact that its intensity is high.

6-      On page 14, line 245-247, the sentence “Extending our analysis to stage III tumors, we found that all five aptamers showed recognition potential, with better results for AptaB5, AptaB2 and AptaB4, reaching more than 50% recognition (Table 4).” I believe that instead of AptaB2 it should be Apta B1 according to the data presented in table 4. Furthermore, the number of samples used in the recognition stage I and III is low to establish an efficient recognition pattern.

7-      On page 14, line 250, the sentence “with AptaB4 and AptaB2 recognizing most of the cases, and these aptamers were also the most efficient to recognize samples from grades II and III” is not well sustained since AptaB5 (11/29 stage II and 5/9 stage III) has better efficiency in the recognition of grade II samples than AptaB2 (10/29) and in grade III AptaB5 also has better recognition efficiency than AptaB2 (4/9) and AptaB4 (4/9).

8-      The discussion in the structural characterization has to be implemented. The conclusion achieved by the authors is not clear after the study of the 2D structure carried out using two different servers and its 3D comparison in pymol.

9-      The 2D structures of the aptamers in Figure 9 shoul be improved.

10-  A lack of references is observed throughout the discussion section, it is recommended to review and add the missing reference in this section.

“Breast cancer is a serious public health problem, and the aim of the present work was to identify specific aptamers for this tumor type with potential diagnostic function. Currently, the diagnosis of the disease is made by searching for receptors present in the tumor cell membrane (ER, PR and HER2), which allows the classification of tumors into major  four molecular subtypes (ref). Tumors expressing the hormone receptors ER and PR or the growth factor HER2 benefit from targeted therapies, and the triple-negative subtype is treated with cytotoxic chemotherapy and, in some cases, immunotherapy (ref). However, in  clinical practice, the molecular classification based on these three receptors is not sufficient to define a more accurate diagnosis and prognosis, mainly to the triple negative subtype (ref). The study of ssDNA aptamers capable of detecting the disease, can serve as a tool to complement the current used diagnosis, and the identification of the aptamer’s targets could be used as new TNBC biomarkers, thus developing strategies for personalized therapy.......

Author Response

Rio de Janeiro, October 30th 2023

Dear Reviewer,

We are very pleased with your report concerning our manuscript “Aptamer-based recognition of breast tumor cells: a new era for breast cancer diagnosis”. Please find below a point-by-point response to the reviewer’s comments.

Reviewer 1 comments.

The manuscript ijms-2674455 entitled “Aptamer-based recognition of breast tumor cells: a new era for breast cancer diagnosis” has identified 5 aptamers, which selectively recognice breast cancer cells, specifically MDA-MB-231 and they do not bind to non-tumoral cell. Aptamers validation studies showed that aptaB4 and aptaB5 have specific recognition for primary and metastatic tissues, respectively. Moreover, the recognition study was extended to other breast cancer subtypes and more complex culture models, such as 3D cultures. Finally, computational biological studies were performed to identify the potential target protein of each one of the aptamers. The results indicate that the target of AptaB4 seems to be CSKP.

There are suggestions for majors’ corrections.

1-      The size of the letters in the legend of the graphic representation of the mean percentage in Figure 1 and 2 should be improved.

Answer: We agree and improved the figures accordingly.

2-      On page 3, line 101, the concentration units (nM) are missing from the Kd value for AptaB2. Furthermore, the Kd values described on this same page and in figure 3 for AptaB2 (206 nM) and for AptaB5 (126 nM) do not match the values presented in the discussion, page 27, line 496.

Answer: We agree and changed the text accordingly. Page 3, line 101 and Page 32, lines 539 and 540.

3-      The replicates and number of assays carried out to obtain the binding values, the Kd and detection capacity of the five aptamers should be indicated.

Answer: We agree and included the information of the number of assays carried out for the binding values in the Materials and Methods section; Analysis of the dissociation constant (Kd), in which we included the sentence: “Three independent replicates were carried out for each test.” Page 37, line 760.

4-      Taking into account the low recognition of non-tumor breast cells and the different pattern in the graph of the dissociation constants (Kd) of the aptamers compared with tumoral cells the values of the Kd shown are misleading. These values of the constants must be removed.

Answer: We agree and removed the Kd values obtained for the non-tumoral MCF-10a cells in the text. Figure 3 was altered to Table 1 in Page 3, line 105.

5-      The sentence “all five aptamers recognized the luminal subtype with high intensity” needs to be more elaborated, considering that the recognition percentage for the Luminal subtype is 3.5% for the aptaB3 despite the fact that its intensity is high.

Answer: We appreciate reviewer consideration and changed the text accordingly: “We observed that the aptamers recognized the luminal subtype with high intensity: AptaB1 (10%), AptaB2 (28%), AptaB4 (42%), and AptaB5 (32%), except for AptaB3 which recognized just one sample (Figure 7, table 2).” in Page 16, line 235.

6-      On page 14, line 245-247, the sentence “Extending our analysis to stage III tumors, we found that all five aptamers showed recognition potential, with better results for AptaB5, AptaB2 and AptaB4, reaching more than 50% recognition (Table 4).” I believe that instead of AptaB2 it should be Apta B1 according to the data presented in table 4. Furthermore, the number of samples used in the recognition stage I and III is low to establish an efficient recognition pattern.

Answer: We agree and changed the text accordingly, including AptaB1 instead of AptaB2 in Page 17, line 260. And the text was changed to “… , although the low number of samples from those stages…” in Page 17, line 260.

7-      On page 14, line 250, the sentence “with AptaB4 and AptaB2 recognizing most of the cases, and these aptamers were also the most efficient to recognize samples from grades II and III” is not well sustained since AptaB5 (11/29 stage II and 5/9 stage III) has better efficiency in the recognition of grade II samples than AptaB2 (10/29) and in grade III AptaB5 also has better recognition efficiency than AptaB2 (4/9) and AptaB4 (4/9). 

Answer: We agree and changed the text accordingly, including AptaB5 instead of AptaB2 in Page 17, line 264.

8-      The discussion in the structural characterization has to be implemented. The conclusion achieved by the authors is not clear after the study of the 2D structure carried out using two different servers and its 3D comparison in pymol.

Answer: Following your recommendation, we have expanded the discussion of the 2D and 3D structural comparisons between the models generated by mFold and NUPACK.
Specifically, we now include an analysis of the predicted free energy values for the 2D folds, which showed some differences between mFold and NUPACK for most aptamers. The text was included in Page 19, line 309.

We also performed a more robust 3D structural alignment using RNA-align, which showed significant differences between the mFold and NUPACK 3D models for most aptamers, except for AptaB2. Figure 9 has been updated to reflect this new structural analysis. Finally, based on the lower minimum free energies and greater 3D folding of the NUPACK models, these were selected as the starting conformations for further refinement using molecular dynamics simulations. Figure 9 was altered in Page 23, line 354 and the correspondent legend in line 355.

9-      The 2D structures of the aptamers in Figure 9 should be improved.

Answer: We agree and improved the figure accordingly. New Figure 8 (old Fig. 9) was altered in Page 21, line 348.

10- A lack of references is observed throughout the discussion section, it is recommended to review and add the missing reference in this section.

“Breast cancer is a serious public health problem, and the aim of the present work was to identify specific aptamers for this tumor type with potential diagnostic function. Currently, the diagnosis of the disease is made by searching for receptors present in the tumor cell membrane (ER, PR and HER2), which allows the classification of tumors into major four molecular subtypes (ref). Tumors expressing the hormone receptors ER and PR or the growth factor HER2 benefit from targeted therapies, and the triple-negative subtype is treated with cytotoxic chemotherapy and, in some cases, immunotherapy (ref).41,42 However, in clinical practice, the molecular classification based on these three receptors is not sufficient to define a more accurate diagnosis and prognosis, mainly to the triple negative subtype (ref). 43 The study of ssDNA aptamers capable of detecting the disease, can serve as a tool to complement the current used diagnosis, and the identification of the aptamer’s targets could be used as new TNBC biomarkers, thus developing strategies for personalized therapy.......

Answer: We agree and included references 41, 42 and 43 in the discussion section in the Page 31, lines 513 and 515.

Reviewer 2 Report

Comments and Suggestions for Authors

In the paper, the authors attempted to identify aptamers that is specific for breast cancer. However, several deficiencies were noted:

1) Other studies have used the same cell-SELEX on the same cell line (MDA-MB-231), e.g. PMID 33883615. Moreover, the aptamers they identified have even higher affinity than those found in this work. This makes me question the value of the current study.

2) In addition, although the authors intend to identify markers for triple negative breast cancer (MDA-MB-231), they ended up having a marker for all breast cancer subtypes with poor specificity. In this case, they are not good markers for patient diagonsis. 

3) Unvalidated docking data based on computational analysis is not reliable and must be verified with experimental evidence. 

Comments on the Quality of English Language

The quality of english is fine.

Author Response

Rio de Janeiro, October 30th 2023

Dear Reviewer,

We are very pleased with your report concerning our manuscript “Aptamer-based recognition of breast tumor cells: a new era for breast cancer diagnosis”. Please find below a point-by-point response to the reviewer’s comments.

Reviewer 2 comments.

In the paper, the authors attempted to identify aptamers that is specific for breast cancer. However, several deficiencies were noted:

1) Other studies have used the same cell-SELEX on the same cell line (MDA-MB-231), e.g. PMID 33883615. Moreover, the aptamers they identified have even higher affinity than those found in this work. This makes me question the value of the current study.

Answer: We appreciate the reviewer`s point of discussion and agree that a variety of studies were performed to select aptamers targeting breast cancer cells (PMID: 30098503, PMID: 32222697, PMID: 24892693, PMID: 26730812, PMID: 37680989, PMID: 37363395). Nevertheless, the fact that other groups have also the same interest as we do is not an excluding point. On the contrary, it adds more confidence to the use of aptamers as a new diagnostic and therapeutic tools in the near future. Moreover, it brings the possibility to improve the methodology by changing ideas and protocols. Indeed, the study performed by Ferreira et al., 2021 showed aptamers with specificity for the MDA-MB231 cells with high affinity, better than ours, but our data presents a deeper analysis showing its application in clinical samples (n=100, including tumoral and non-tumoral adjacent tissues), which brings robustness and relevance to the study.

2) In addition, although the authors intend to identify markers for triple negative breast cancer (MDA-MB-231), they ended up having a marker for all breast cancer subtypes with poor specificity. In this case, they are not good markers for patient diagnosis. 

Answer: We understand the reviewer’s point of view, but in our opinion the fact that the set of aptamers selected by us being able to detect all breast cancer subtypes is not a problem. On the other hand, it could be used as a high range detection of breast cancer, including cases of early stage, which could be applicable for early detection. Moreover, currently, the triple negative diagnosis is based by the exclusion of hormone receptors and/or HER2 expression and our TMA analysis showed important data of the recognition of primary and metastatic sites from triple negative molecular subtypes, indicating that the diagnosis for this subtype would not be by exclusion. Therefore, the aptamers panel described here could improve breast cancer diagnosis for all subtypes, including the early stages and metastatic sites.

3) Unvalidated docking data based on computational analysis is not reliable and must be verified with experimental evidence. 

Answer: We completely agree with the reviewer’s point of view of the importance of validating the proteins as potential targets to the selected aptamers. Nevertheless, this validation is beyond the scope of the current work and will be performed in our laboratory very soon. But we agree that this is an important argument and thus there is a sentence in the discussion section: “Thus, it would be important to further validate this data using bench approaches, such as surface plasmon resonance and calorimetry methods.” Page 34, line 666.

Reviewer 3 Report

Comments and Suggestions for Authors

In this work, the authors selected five new aptamers for the detection of breast tumour cells. They used different techniques, ranging from biological to computational analysis. The work, well organized and written, is interesting and offer a valuable contribution to the anticancer field, thus describing an alternative and valuable approach for breast cancer diagnostic. 

In my opinion, this paper could be of interest for the typical readership of Int.J.Mol.Sci. However, a few issues should be considered before publication.

1)    My first concern is about the Kd analysis showed in section 2.1. In my opinion, the performed analysis is not accurate. First of all, the number of the plotted points is not sufficient considering the huge errors associated to some measurements. Then, for a correct Kd extrapolation, the X=0,  Y=0 should always be indicated in the plotting. Moreover, the errors associated to the Kd values should always be reported.

2)    The computational analysis is very interesting and give important information about aptamer conformations and potential targets in cells. However, without supporting experiments, these information are no more that approximative indications of what really happen in cells. The authors obtained good in silico results with CSKP-AptaB4 complex. In my opinion, the authors should validate at least the latter result to take advantage of all the work they did.

A few typos are present in the text and should be fixed. Among others, see:

a)     Section 2.1, line 98: there is an underscore between “the” and “binding”;

b)    Section 2.5, line 191:  “aplication” should be “application”.

Comments on the Quality of English Language

Author Response

Rio de Janeiro, October 30th 2023

Dear Reviewer,

We are very pleased with your report concerning our manuscript “Aptamer-based recognition of breast tumor cells: a new era for breast cancer diagnosis”. Please find below a point-by-point response to the reviewer’s comments.

Reviewer 3 comments.

In this work, the authors selected five new aptamers for the detection of breast tumour cells. They used different techniques, ranging from biological to computational analysis. The work, well organized and written, is interesting and offer a valuable contribution to the anticancer field, thus describing an alternative and valuable approach for breast cancer diagnostic.

In my opinion, this paper could be of interest for the typical readership of Int.J.Mol.Sci. However, a few issues should be considered before publication.

  • My first concern is about the Kd analysis showed in section 2.1. In my opinion, the performed analysis is not accurate. First of all, the number of the plotted points is not sufficient considering the huge errors associated to some measurements. Then, for a correct Kd extrapolation, the X=0, Y=0 should always be indicated in the plotting. Moreover, the errors associated to the Kd values should always be reported.

Answer: We completely agree with the reviewer`s point of view about the low number of the plotted points and the errors associated to some measurements. As we used a single point of the control aptamer library with the highest concentration of 400nM, we could not show the X=0 and Y=0 in the plot as suggested. In order to present a clear data, we decided to change the graph Kd result to a Table informing each Kd obtained (new Table 1) and Figure 3 was excluded. Page 3, line 105.

  • The computational analysis is very interesting and give important information about aptamer conformations and potential targets in cells. However, without supporting experiments, these information are no more that approximative indications of what really happen in cells. The authors obtained good in silico results with CSKP-AptaB4 complex. In my opinion, the authors should validate at least the latter result to take advantage of all the work they did.

Answer: We completely agree with the reviewer`s point of view of the importance of validating the CSKP as potential target to the aptamer AptaB4. Nevertheless, this validation is beyond the scope of the current work and will be performed in our laboratory very soon. But we agree that this is an important argument and thus there is a sentence in the discussion section: “Thus, it would be important to further validate this data using bench approaches, such as surface plasmon resonance and calorimetry methods.” Page 34, line 666.

A few typos are present in the text and should be fixed. Among others, see:

  1. Section 2.1, line 98: there is an underscore between “the” and “binding”;

Answer: The text was corrected accordingly. Page 2, line 98.

  1. Section 2.5, line 191: “aplication” should be “application”.

Answer: The text was corrected accordingly. Page 14, line 203.

Reviewer 4 Report

Comments and Suggestions for Authors

In this manuscript, the authors identified aptamers specific to MDA-MB-231 tumor cell line using Cell-SELEX technique. Five aptamers were selected as binding with breast cancer cell lines but not with non-tumor breast cells. These aptamers could recognize primary and metastatic tumors of all subtypes. Furthermore, they identified CSKP as a potential target that is interacting with AptaB4. These findings indicate that aptamers hold promise for breast cancer diagnosis and treatment, thanks to their specificity and selectivity.

1. The authors should add a section where they can discuss the cell SELEX procedure in detail. I would suggest using a figure to explain in detail.

2. They should discuss more on Figure 3 regarding the dissociation constant of aptamers binding with non-tumor control MCF-10A. Does the dissociation constants difference big enough to be used in real world application?

3. In Figure 4, they discussed the distribution of subcellular localization of the aptamers. What is the main purpose of knowing the distribution of the aptamers? How many cells did they analyzed to obtain those percentage numbers?

4. There is some inconsistency in the Figure 4. Aptamer 2 and 3 have been shown to nonspecifically bind with non-tumor cells. While in Figure 4B, none of these two aptamers could recognize the non-tumor cells. They should explain more on this.

5. On line 146, they stated that no labeling was observed for non-tumoral control cell line MCF-10A. While this is not true. You could still see minor level of labeling here. Similar issue happened on line 166 as well.

6. Based on data from Table 7, even if you combine all five aptamers, the specificity is still really low. Is there other method to solve this issue? Maybe by optimizing cell SELEX procedure using more rounds of negative SELEX?

Comments on the Quality of English Language

Moderate changes required

Author Response

Rio de Janeiro, October 30th 2023

Dear Reviewer,

We are very pleased with your report concerning our manuscript “Aptamer-based recognition of breast tumor cells: a new era for breast cancer diagnosis”. Please find below a point-by-point response to the reviewer’s comments.

Reviewer 4 comments.

In this manuscript, the authors identified aptamers specific to MDA-MB-231 tumor cell line using Cell-SELEX technique. Five aptamers were selected as binding with breast cancer cell lines but not with non-tumor breast cells. These aptamers could recognize primary and metastatic tumors of all subtypes. Furthermore, they identified CSKP as a potential target that is interacting with AptaB4. These findings indicate that aptamers hold promise for breast cancer diagnosis and treatment, thanks to their specificity and selectivity.

1.The authors should add a section where they can discuss the cell SELEX procedure in detail. I would suggest using a figure to explain in detail.

Answer: In the Materials and Methods section there is a topic explaining the Cell SELEX method used in the present work. But we agree that the addition of a scheme explaining the procedure in more details would improve readability. Therefore, we included a scheme characterizing the cell SELEX procedure used in the present work, Figure 15 in the Page 36, line 711.

  1. They should discuss more on Figure 3 regarding the dissociation constant of aptamers binding with non-tumor control MCF-10A. Does the dissociation constants difference big enough to be used in real world application?

Answer: The data obtained for Kd values of MCF-10A cells were also questioned by the other reviewers, which suggested to remove it. Therefore, we removed the Kd values obtained for the non-tumoral MCF-10a cells in the text. The absence of detection of non-tumoral breast cells was also confirmed by aptafluorescence assay using 2D and 3D culture cells models.

  1. In Figure 4, they discussed the distribution of subcellular localization of the aptamers. What is the main purpose of knowing the distribution of the aptamers? How many cells did they analyzed to obtain those percentage numbers?

Answer: In the aptamer field based on cell SELEX selection method, many intriguing questions arises, such as: which could be the aptamers target in the cell; if after binding to its target, the aptamer could address some cellular function in the cell; if there is aptamer internalization, and if the aptamers are degraded after internalization. To try to answer these questions, we performed an analysis of the aptamer intracellular localization after 1 hour of incubation with each individual aptamer, by counting 300 cells. This point is already discussed in Page 32, line 570.

  1. There is some inconsistency in the Figure 4. Aptamer 2 and 3 have been shown to nonspecifically bind with non-tumor cells. While in Figure 4B, none of these two aptamers could recognize the non-tumor cells. They should explain more on this.

Answer: We appreciate reviewer comment, and we suppose it is referring to a comparison between Figures 2 and new Figure 3. Indeed, as observed in Figure 3, by the use of the 2D culture model, no aptamer binding was observed for the non-tumoral breast cell MCF-10A, but the flow cytometry analysis revealed MCF-10A recognition by AptaB2 and AptaB5 as observed in Figure 2. Therefore, we could speculate that the trypsinization process performed for the flow cytometry analysis could altered cell membrane and thus may have caused this inconsistence, as opposed to the adhered cells used in aptafluorescence. Then, we performed a third method to confirm the specificity of the aptamers: the 3D culture model, in which we have noticed a very weak staining in the non-tumoral cells and thus, after comparing both staining intensity and binding proportional to non-tumoral and tumoral cells, we have considered that the aptamers were specific for breast tumor cells.

We included the sentence in the discussion section: “Curiously, by the use of the 2D culture model, no aptamer binding was observed for the non-tumoral breast cell MCF-10A, but the flow cytometry analysis revealed MCF-10A recognition by AptaB2 and AptaB5. Therefore, we could speculate that the trypsinization process performed for the flow cytometry analysis could altered cell membrane and thus may have caused this inconsistence, as opposed to the adhered cells used in aptafluorescence.” in Page 32, line 564.

  1. On line 146, they stated that no labeling was observed for non-tumoral control cell line MCF-10A. While this is not true. You could still see minor level of labeling here. Similar issue happened on line 166 as well.

Answer: We agree and changed the text as follows: “In parallel, to validate the specificity of aptamer recognition for tumor cells only, we also constructed a 3D model of the non-tumoral control cell line MCF-10A, and an absence or a very weak labeling was observed (new Figure 4B-F).” in Page 11, line 157.

  1. Based on data from Table 7, even if you combine all five aptamers, the specificity is still really low. Is there other method to solve this issue? Maybe by optimizing cell SELEX procedure using more rounds of negative SELEX?

Answer: We appreciate the reviewer’s point of discussion and agree that the specificity observed in our TMA analysis was still low, even combining the five aptamers. We speculate that this issue is caused by the type of tissue used as non-tumoral control ones, which was included in the TMA block purchased by us. Actually, the control samples were tissues adjacent to the tumor, and we could not exclude the possibility of initial grades of biomarker presence caused by the tumor microenvironment, which could be detected by the aptamers. To solve this issue, further analysis should be performed using truthful non-tumor breast tissue, arising from mammoplasty, for example.  This point was already discussed in Page 34, line 628.

Round 2

Reviewer 1 Report

Comments and Suggestions for Authors

The authors essentially addressed all the major issues raised by the Reviewer. I have no additional concern and in my opinion this article can be published as is.

Reviewer 2 Report

Comments and Suggestions for Authors

The authors have addressed my comments. 

Comments on the Quality of English Language

No issues.